# Optimizing hyperparameters of YOLOv10 for arson detection using advanced optimization algorithms

Ali Abbas Abbod [1,2*], Matheel E. Abdulmunim[1], Ismail A. Mageed[3]

**1** Computer Sciences college, University of Technology-Iraq, Baghdad, Iraq, **2** Department of Computer Technology Engineering, Technical College, Imam Ja'afar Al-Sadiq University, Iraq, **3** Institute of Education, Sheffield Hallam University, United Kingdom

* cs.22.19@grad.uotechnology.edu.iq

## Abstract

Arson detection plays a critical role in protecting lives and property in high-risk environments such as airports, industrial zones, and other public areas. Recent advances in deep learning (DL), particularly YOLOv10, have demonstrated strong potential in real-time object detection. However, the model's performance is highly dependent on effective hyperparameter optimization to maintain a balance between accuracy and computational efficiency. This study proposes a hybrid GWO-BBOA optimization algorithm that combines the global search capability of Grey Wolf Optimization with the fine-tuning strength of the Brown Bear Optimization Algorithm to optimize YOLOv10 for arson detection. The model was evaluated using an augmented dataset of 2,182 annotated images. Experimental results show that the proposed GWO-BBOA approach outperforms traditional optimization methods, achieving a recall of 0.620. This indicates its enhanced ability to detect true arson events. Moreover, the hybrid algorithm effectively balances exploration and exploitation while reducing the number of required iterations. Future work will focus on expanding the dataset, implementing adaptive optimization strategies, and integrating the model into real-time surveillance systems. Overall, this work highlights the value of hybrid metaheuristic approaches in enhancing DL models for safety-critical applications.

## 1. Introduction

The improvements in deep learning (DL) technology are proving worthy in high-risk areas such as centers, airports, train stations, manufacturing plants, and other critical inhabited structures where continuous monitoring is paramount for the protection of lives and property [1]. In the case of DL, it is feasible to identify abnormalities and handle huge volumes of videotape data [2,3]. Among these abnormalities, is arson, which is predicted to threaten some societies because it causes property damage

**Data availability statement:** The minimal anonymized dataset necessary to replicate the findings of this study has been deposited at Kaggle: https://www.kaggle.com/datasets/aliab-basabbod/arson-dataset The source code used to train aBefore we proceed, please disclose whether or not the participant shown in Figure XX consented to having this image published under the Creative Commons Attribution (CC BY) license and signed the PLOS Consent Form for Publication in a PLOS Journal (https://journals.plos.org/plosone/s/file?id=8ce6/plos-consent-form-english.pdf). If the participant completed the consent form, please amend the methods section and ethics statement of the manuscript to explicitly state that the patient/participant has provided consent for publication: "The individual pictured in Fig _____ has provided written informed consent (as outlined in PLOS consent form) to publish their image alongside the manuscript". If the participant did not complete this consent form, please remove this image from the figure, as we cannot publish the image without the signed PLOS Consent Form. nd evaluate the models is available at GitHub: https://github.com/AliAbbasAbbod/YOLOv10-Arson-Optimization.

**Funding:** The author(s) received no specific funding for this work.

**Competing interests:** The authors have declared that no competing interests exist.

and loss of life [4]. Early detection of damages reduces loss of life, compliance with authorities and preservation of the environment [5].

High-performance object detection model YOLOv10 is renowned for its accuracy and effectiveness, especially in difficult situations like dimly lit areas. Advanced features like the removal of Non-Maximal Suppression (NMS), the incorporation of Partial Self-Attention (PSA), and a brand-new Dual Label Assignment approach that facilitates both one-to-many and one-to-one matching are introduced. Real-time context awareness and detection accuracy are greatly increased by these improvements. YOLOv10 is also very portable and consistently performs well on a variety of platforms, including mobile devices and automotive systems. This makes it perfect for surveillance situations where accuracy, speed, and adaptability must be balanced [6,7].

Nevertheless, the effectiveness of such models greatly depends on the values of hyperparameters that define them, which greatly affect both accuracy and time to complete a query. Tuning the parameters, including learning rate, batch size, and epochs, ensures that models like YOLOv10 detect anomalous circumstances in the quickest way with precise and accurate action that can hardly go wrong [8]. Our work assesses the effectiveness of an arson detection system based on the YOLOv10 model that is fine-tuned with the hybrid GWO-BBOA method. GWO as a metaheuristic is modeled on hunting mechanisms of grey wolves operating in packs and is subdivided into four types of wolves: alpha, beta, delta, and omega, targeting the roles of hunting, searching for prey, and optimizing the search space. This structure helps in developing structure search strategies for the identification of highly promising parameters and in creating a compromise between local and global methods [9]. However, the Brown-Bear Optimization Algorithm (BBOA) mimics the foraging strategy used by brown bears and their pedal signs. This dynamic approach permits several optimization phases, including gait walking as well as sniffing pedal marks to search new zones in an efficient and optimized method. By such bifurcated methods, the BBOA offers optimal and moderate hyperparameters for YOLOv10 [10]. Thus, when the organized leadership strategy of GWO is combined with the dynamic behavior of BBOA, the result is, a high level of efficiency. It can improve security measures during sensitive and risky moments.

Traditional metaheuristic optimization techniques like PSO and GA, as well as earlier object detection techniques like YOLOv9, frequently falter in dynamic and noisy surveillance settings. Premature convergence, poor generalization in dimly lit or obscured scenes, and inadequate hyperparameter tuning are some of these drawbacks [11]. In order to improve robustness, accuracy, and parameter search efficiency for arson detection tasks, this study suggests a novel framework that combines YOLOv10 with a hybrid GWO-BBOA optimization strategy.

*The main contributions of this paper are as follows:*

1. *Optimizing the YOLOv10 model for arson detection using the proposed hybrid GWO-BBOA algorithm.*

2. *Comparing the results of a new optimization method called hybrid GWO-BBOA with other optimization methods, including PSO, GWO, and BBOA, to prove the efficiency of our proposed method in enhancing the YOLOv10 performance.*

*The remainder of this paper is organized as follows: Section 2 reviews relevant previous studies related to anomaly (arson) detection and optimization algorithms. Section 3 presents the proposed methodology, including the structure of the YOLOv10 model and the hybrid GWO-BBOA optimization strategy. Section 4 experimental setup and discusses the results and performance comparison with other optimization methods. Finally, Section 5 concludes the paper and outlines directions for future research.*

## 2. Previous studies

Previous studies have shown the huge benefit of anomaly detection in improving the accuracy of various deep learning models as well as optimization algorithms, For example, [12] presented rapid determination of whether an unexpected action is odd or suspicious. It is necessary to indicate which frames and segments of the recording feature the unusual activity. Thus, they used deep learning techniques to automate the threat recognition system to minimize the waste of labor and time. It aims to automatically distinguish abnormalities from typical patterns by recognizing aggressive and violent indicators in real time. They plan to apply two distinct deep learning models (CNN and RNN) to detect and categorize high movement levels in the frame. Subsequently, they were able to detect warnings in the event of a threat, highlighting any suspicious activity at that particular moment. In the context of optimization algorithms, [13] considered the use of PSO for DL models using activation functions and batch size. Another advantage of the PSO approach is the ability of swarm-based strategy for searching for the solution space with a reduced chance of convergence to local optima, hence enhancing model accuracy. In contrast, [14] developed GWO (Grey Wolf Optimizer) based on the leadership and hunting mechanisms of real-life grey wolves. The proposed GWO approach offers a positive example of exploring and exploiting experience as demonstrated by the alpha, beta, and delta wolves, which also demonstrates its ability to optimize hyperparameters to address complex problems, including classification and regression. On the other hand, [15] highlighted the idea of BBOA, which is a new approach for dynamic optimization in learning phases and adjusts learning rates for model stability applications like classification and segmentation. Altogether, all these algorithms show special versatility in enhancing the performance, speed, and flexibility of DL frameworks.

The use of a hybrid of multiple optimization algorithms in hyperparameter optimization has been a success in enhancing the overall performance of the model. Furthermore, [16] proposed the integration of GA and XGBoost in fraud detection tasks, with good gains in terms of efficiency and accuracy. This approach highlighted the advantage of combining metaheuristic approaches in handling constraints that may be inherent in them when used singularly.

Finally, [17] evaluated the performance of the YOLOv9 model for detecting arson incidents in surveillance videos. The study demonstrated that while the model achieved a precision of 0.552 overall in arson classification, it could reach a confidence level of 0.933 when predicting certain events. These findings confirm the feasibility of using YOLO-based architectures for arson detection, although further enhancements in model tuning and environmental robustness were suggested for broader applicability. This study forms a baseline for the current work, which aims to improve detection accuracy using the YOLOv10 model with advanced hybrid optimization.

These studies highlight the increasing relevance of integrated metaheuristic approaches for improving the predictive outcomes. Nevertheless, the application of GWO and BBOA in optimizing the state-of-art models such as YOLOv10 has little attention in the literature, such research will fill the gap in the existing knowledge and contribute significantly to the advancements in the field.

## 3. Methodology

In this section, the authors describe the general approach for identifying arson, the approach to data collection, and the training and validation of the models. The proposed approach experiments with complex methods for hyperparameter tuning to get the best result from the YOLOv10 model, as shown in Fig 1. In this regard, PSO, GWO and BBOA methods are proposed and implemented to solve the problem. Besides, as a result of the potential inability of one single algorithm to provide improved detection accuracy for this model, a novel meta-heuristic algorithm called GWO_BBOA is introduced.

## 3.1 Data preparation and dataset exploration

Preparation procedures in an effort to incorporate robustness included cropping out details in images and standardizing the images to 640 x 640 pixels from the Anomaly Detection Dataset (https://www.kaggle.com/datasets/aliabbasabbod/arson-dataset) that collected in UNC Charlotte, where 290 frames were manually selected from 53 videos that had arson scenes. Since the number of selected images is minimal, which directly affects the performance of the model, data augmentation are used with the following transformation techniques. These transformations are included: rotation (up to ±45°), horizontal flipping (100% probability), random scaling (±20% of the original size), brightness and contrast adjustments (±20% variation), Gaussian noise addition (variance range: 10–50), Gaussian blurring (blur limit of 3), contrast enhancement (±20% variation), gamma correction (range: 80–120), and hue-saturation adjustments. These transformations ensured variability in the dataset by simulating different lighting conditions, perspectives, and image distortions. As a result, the total number of images increased to 2,182. While we use data augmentation techniques to make the dataset appear larger, they are not a complete replacement for diverse, real-world samples. Therefore, the model's capacity to generalize to new arson scenarios and environmental conditions may be hampered by the lack of truly independent frames. Furthermore, the researcher then used a Python script to split the data into three groups: a 70% training set, a 20% validation set, and a 10% test set. Following that step, the process involved annotating the images and creating bounding boxes through the labelingImg program [18]. These annotations were crucial in defining the boundaries of each image for analysis and comprehension. This process guarantees that the model can accurately recognize components in cases of arson.

The YOLOv10 model takes in RGB image frames that have been resized to a set resolution of 640×640 pixels. We normalize each frame by scaling the pixel values to the range [0, 1], and we keep the three-color channels (RGB). For inference, these preprocessed frames are sent to the model at a rate of 25 frames per second. The model's output is a list of bounding box predictions in the format [class label, x_min, y_min, x_max, y_max, confidence score]. We don't look at predictions with a confidence score lower than 0.25. YOLOv10 doesn't need external Non-Maximum Suppression (NMS) like earlier versions did because it has a built-in Dual Label Assignment mechanism. If the final detections show people interacting with flammable materials, fire activity, or suspicious postures within the defined bounding boxes, they are considered arson events.

## 3.2 Model selection and architecture

YOLOv10 became the focus of this study due to its high performance and reliability in tasks associated with identifying objects quickly and effectively in complex cases such as arson and fires. Compared to previous versions like YOLOv8, which is faster but less accurate in complex scenes, slower models like Faster R-CNN, YOLOv10 is a perfect balance between speed and accuracy to respond real-time in complex scenarios [19–22]. This latest model introduces a training method that removes the requirement for NMS by employing the dual label assignment concept that combines the advantages of both "one-to-many" and "one-to-one" matching strategies. The "one-to-many" technique provides hints and suggestions while the "one-to-one" strategy eliminates the need for NMS in inference procedures to improve efficiency. Through training these approaches, YOLOv10 ensures predictions without compromising on performance quality. The reliable matching metric ensures that both heads are aligned in supervision to ensure that the "one-to-one" head gets the guidance offered by the "one-to-many" head, as shown in Fig 2. YOLOv10 also brings in enhancements, like Compact Inverted Blocks (CIB) and PSA, which help lessen the computational workload. These advancements lower delays by 46% and cut down on parameters by 25% when contrasted with versions such as YOlOv8 but still deliver the same level of excellence [23].

These enhancements therefore provide the theoretical support to the effectiveness of the YOLOv10 in the given arson detection task.

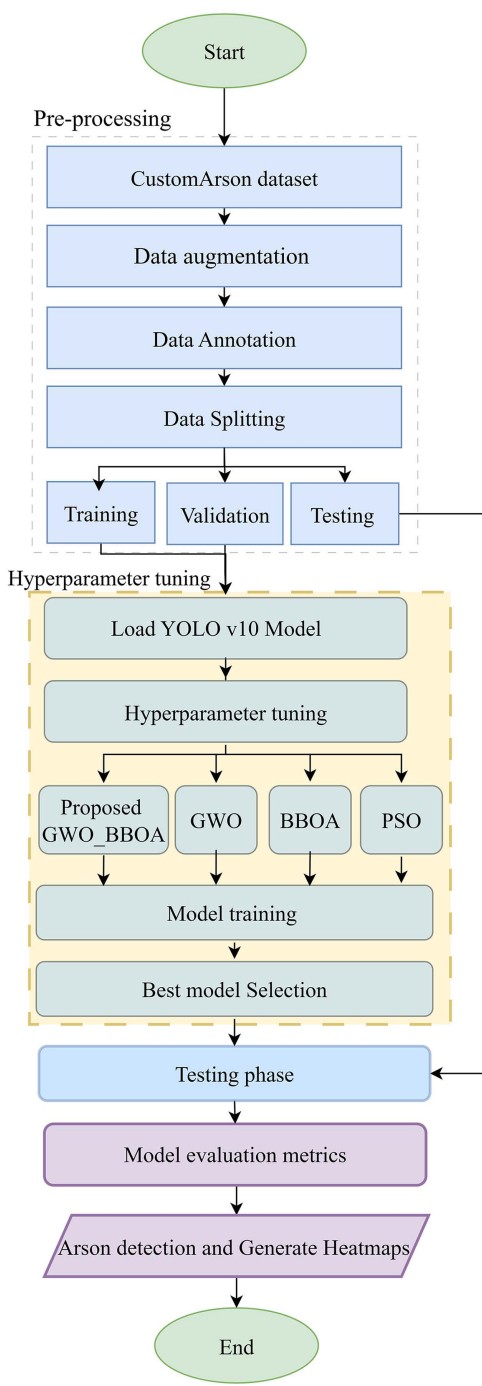

**Fig 1. A flowchart for methodology.**

## 3.3 Optimization algorithms for hyperparameter tuning

In modern ML and DL sets, hyperparameters are important in determining model performance. Hyperparameters such as learning rate (lr0), Learning Rate Factor (lrf), Momentum (mo) and weight decay (wd) enhance quick

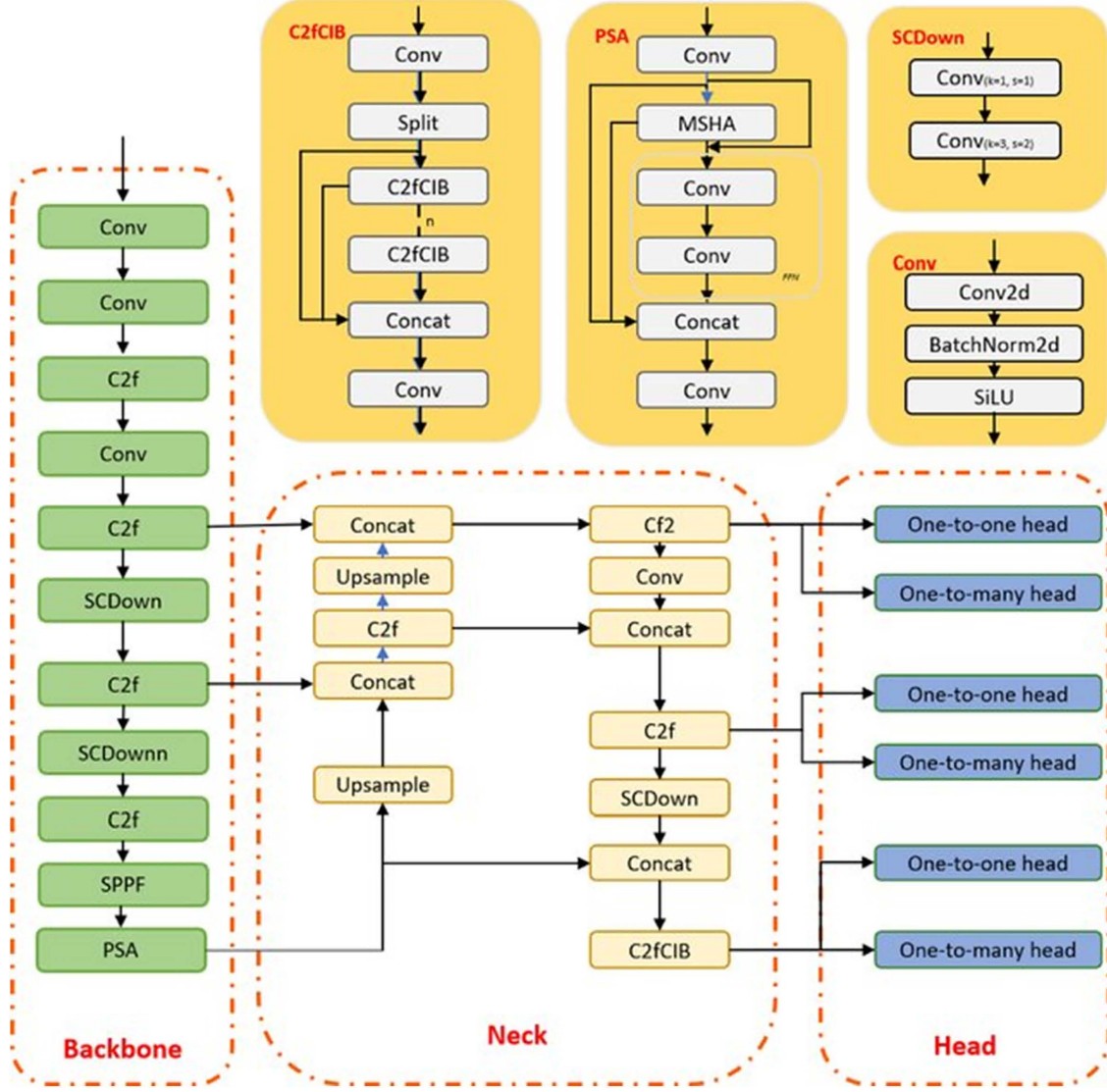

**Fig. 2. Architecture of YOLOv10 [24].**

convergence beside ensuring good generalization capacity that greatly reduces other risks such as overfitting or underfitting [25]. Balancing these parameters is most important for a task when a certain set of parameters must exceed certain values; for example, for object detection in conditions of active movement arson detection. A rapid response is essential in such cases, making precision a top priority. The primary difficulties regarding hyperparameter tuning have been solved by growing more sophisticated approaches which draw on the natural and evolutional procedures. These optimizations utilize various approaches in the appropriate searching for solutions within a solution space efficiently, with an aim of selecting hyperparameters that will enhance model performance while at the same time reducing the costs of computation. Manual tuning is often time-consuming and inefficient. Although techniques like random adjustment exist, their effectiveness varies significantly depending on environmental conditions [26]. These hyperparameters were carefully chosen according to the literature on DL, including their significance and

possible influence [27]. A description and the recommended ranges of key YOLO hyperparameters are presented in Table 1.

Many methods for hyperparameter tuning exist but to execute them fast enough within the space of computational tractability, sophisticated optimization algorithms are required for good exploration and exploitation. In this study, four well-established optimization algorithms were applied alongside a novel hybrid approach: Algorithms used are Particle Swarm Optimization (PSO), Grey Wolf Optimizer (GWO), BBOA, and the improved Hybrid GWO-BBOA (proposed). These algorithms were selected because of their efficiency and also their different properties in searching for the best set of hyperparameters in the solution space [33]. Every time they have the capability to tradeoff between exploitation and exploration, tackle challenging optimization spaces, and improve on model fixation makes them ideal in optimizing DL models such as YOLOv10. In addition, the idea of the proposed hybrid GWO-BBOA attempts to bring the advantage of both GWO and BBOA to solve one of the hyperparameter optimization issues in complicated detection problems, including arson detection. As the following sections provide a detailed exploration of each algorithm, highlighting its mechanism and application in optimizing YOLOv10 hyperparameters.

**3.3.1 Particle swarm optimization.** The basic idea of PSO is to avoid collisions with neighbors, stay near neighbors and match the velocity of neighbors. A collection of particles, each of which represents a potential solution to the optimization issue, makes up the swarm in PSO algorithms. Every particle in the swarm possesses a position and a velocity, which are constantly updated according to the particle's experiences as well as those of the other particles. Every particle's position is updated based on its current velocity. This type of process allows the swarm to find the right hyperparameters for the YOLOv10 model according to Eq. (1,2) [34].

$$v_i(k+1) = w * v_i + c_1 * R_1 * (pbest - x_i(k)) + c_2 * R_2 * (gbest - x_i(k)$$

(1)

$$x_i(k+1) = x_i(k) + v_i(k+1))$$

(2)

**3.3.2 Grey Wolf Optimization.** The basic idea of (GWO) is to use the leadership and teamwork behaviors observed in grey wolves to solve complex optimization problems. Specifically, four types of grey wolves, alpha, beta, delta, and omega, are employed. The algorithm identifies and designates the top three solutions as alpha (α), beta (β), and delta (δ) wolves, which represent the best, second-best, and third-best solutions, respectively. Consequently, these leaders, considered omega (ω) wolves, guide the rest of the pack. The positions of the pack members are adjusted according to the positions of the alpha, beta, and delta wolves as these leading wolves search towards the optimal solution [35], as shown in Fig 3.

**Table 1. Description and recommended ranges of key YOLO hyperparameters.**

| Hyperparameter Name & Reference | Description | Range |
|---|---|---|
| lr0 [28,29] | controls how quickly the model modifies its weights while it is being trained. A greater lr may speed up the model's convergence, but it may also cause the model to miss the ideal weights, which would impair performance. Conversely, a lower (lr) might result in more steady convergence but could also lengthen the time it takes the model to find the optimal solution | [0-1] |
| lrf [30] | is usually used to adjust the (lr) during the training period. This factor allows the (lr) to be changed, helping to achieve a better balance during the learning process and avoid the problem of gradient descent | [0-1] |
| mo [31] | is a technique used to speed up the training process in the right direction and reduce oscillation | [0-1] |
| wd [32] | is a regularization method that targets the size of the model's weights by applying a penalty to the loss function. The weight decay parameter regulates the severity of this penalty. This method can improve the model's performance on new data and prevent overfitting | [0-1] |

Mathematically this is modeled by the following equations. To calculate the distance between a grey wolf and the prey the formula in Equation (3):

$$\vec{D} = \left| \vec{C} \cdot \vec{X}_{p(t)} - \vec{X}(t) \right| \tag{3}$$

The position of a grey wolf is then updated using Equation (4):

$$\vec{X}(t+1) = \vec{X}_{p(t)} - \vec{A} \cdot \vec{D} \tag{4}$$

Where the coefficient vectors $\vec{A}$ and $\vec{C}$ are defined as Equation (5) and Equation (6):

$$\vec{A} = 2\vec{a} \cdot \vec{r}_1 - \vec{a} \tag{5}$$

$$\vec{C} = 2 \cdot \vec{r}_2 \tag{6}$$

**Where:**

• t denotes the current iteration.

• $\vec{X}p$ (t) the position vector of the prey.

• $\vec{X}(t)$ is the position vector of the grey wolf.

• $\vec{D}$ is the distance vector between the grey wolf and the prey.

• $\vec{A}$ *and* $\vec{C}$ are coefficient vectors used to guide the position update.

• $\vec{a}$ is a linearly decreasing vector from 2 to 0 over the course of iterations.

• $\vec{r}_1$ *and* $\vec{r}_2$ are random vectors with components in the range [0,1].

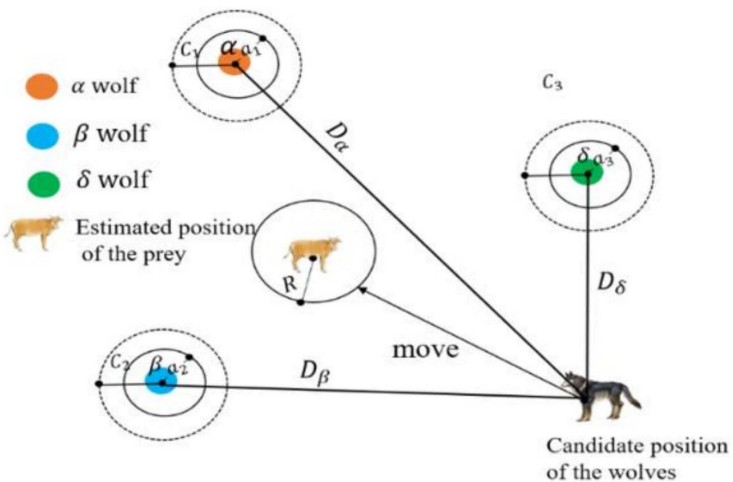

**Fig. 3. GWO Strategy [36].**

During the hunting phase, the wolves update their positions with respect to the best three leaders, according to Equation (7) and Equation (8):

$$\vec{D}_\alpha = \left| \vec{C}_1 . \vec{X}_{\alpha-} \vec{X} \right| = \left| \vec{C}_2 . \vec{X}_{\beta-} \vec{X} \right|, \vec{X}_\delta = \left| \vec{C}_3 . \vec{X}_{\delta-} \vec{X} \right| \tag{7}$$

$$\vec{X}(t+1) = \vec{X}_{p(t)} - \vec{A} \cdot \vec{D} \tag{8}$$

Then, the final position of the wolf is calculated as the average:

$$\vec{X}(t+1) = \frac{\vec{X}_1 + \vec{X}_2 + \vec{X}_3}{3} \tag{9}$$

These equations (3–9) are based on the original mathematical model of the GWO algorithm [37].

This group behavior enables wolves to strike a balance between exploration and exploitation as they optimize. In case the prey ceases to move, which means the optimal solution was identified, or a stopping condition is achieved, the wolves settle on it and attack the solution. One of the main advantages of the GWO algorithm is its ability to converge towards optimal solutions efficiently [38]. This is especially appropriate when it is used to solve optimization problems that are complex like hyperparameter optimization of the YOLOv10 models where GWO should be able to provide the most effective values of the parameters to use.

**3.3.3 Brown bear optimization algorithm.** New algorithm called Brown Bear Optimization Algorithm (BBOA) is derived from the pedal scent marking and sniffing process of brown bears needed for effective communication and management of territories and resources [39]. It successfully alternates between local and global search in the solution space and thus is ideally suited for application to complex optimization tasks such as hyperparameter optimization for YOLO v10. Brown bears exhibit specific kinds of behavior which are properly incorporated in the BBOA. First behavior is pedal scent marking during which one deliberately step, exhibits specific gait or twists its feet to create scent marks. In the algorithm, this behavior is associated with exploration and refinement phases to allow for an exhaustive search of the solution space. This approach helps to achieve diversification among solutions and prevent the population from stagnating at low-quality solutions. The other important behavior is sniffing whereby the bears try to read the scent marks they have left in order to correct their routes. In the algorithm, this is similar using acquired information for selecting and fine-tuning solutions that allow for a sufficient amount of exploration of the solution space and adequate exploitation of the promising areas. Collectively, these behaviors produce a sound and active process of optimization that is tailored to the challenges of problem-solving environments as in Fig 4. The BBOA utilizes a mathematical model based on the peculiarities of brown bear's activity. The bear population and their starting coordinates in the solution space are limited to a set of fixed constraints. Each position corresponds a potential solution in terms of mathematics and stated as to Equation (10) [39].

$$P_{(i,j)} = P_{\min(i,j)} + \lambda \times (P_{\max(i,j)} - P_{\min(i,j)}) \tag{10}$$

Where $P_{i,j}$ stands for the location of a bear in the search space, limited by $P_{\min(i,j)}$ (lower limit) and $P_{\max(i,j)}$ (upper limit), with $\lambda$ being a random factor between [0,1] that guarantees the differentiation of the initial search. If the total number of groups within a territory (i.e., population) is defined by $N_{pop}$ and total number of pedal marks in each group is defined by D then, the solution set P is represented as Equation (11).

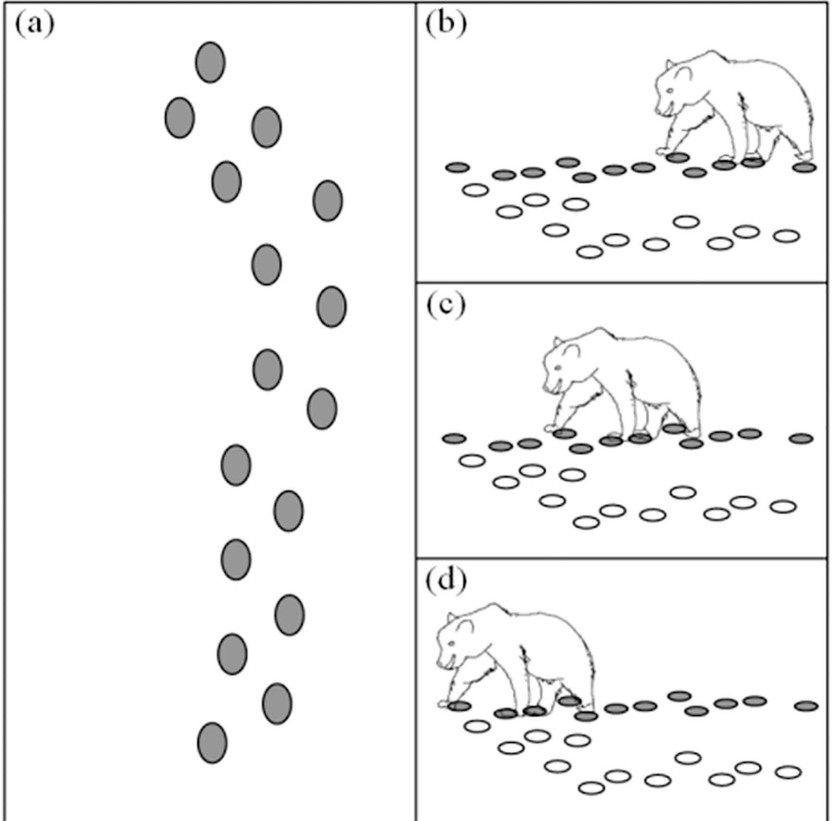

**Fig. 4. Pedal Scent Marking and Sniffing Behaviors of Brown-Bears, (a) Pedal Marks in an Area.** (b-d) A brown-Bear Stretching to its Pedal Marks [39].

$$P = \begin{bmatrix} P_{1,1} & P_{1,2} & \cdots & P_{1,D} \\ P_{2,1} & P_{2,2} & \cdots & P_{2,D} \\ \vdots & \vdots & \ddots & \vdots \\ P_{N_{\text{pop}},1} & P_{N_{\text{pop}},2} & \cdots & P_{N_{\text{pop}},D} \end{bmatrix} \tag{11}$$

Activity of male brown bears is a very important fact within BBOA because the pedal scent marking prevents the searching mechanism directly in a form of distinctive scenting marking. As this behavior is normally seen as performed only by males, each unit in the model is supposed to contain a single male and leave their own mark in terms of scent. These distinctive movement patterns are worked out at the initial exploring stage which is in the initial one-third of the iterations, as follows in the Equation (12).

$$p_{i,j,k}^{new} = p_{i,j,k}^{old} - (\theta_k . \alpha_{i,j,k} . p_{i,j,k}^{old}) \tag{12}$$

where $p_{i,j,k}^{new}$ and $p_{i,j,k}^{old}$ are the updated and previous positions of the $th_j$ pedal mark for the $th_i$ bear group at iteration k, respectively. The parameter $\alpha_{i,j,k}$ is a random number uniformly distributed in [0, 1]. The occurrence factor $\theta_k$, representing the progression of iterations, is given as Equation (13).

$$\theta = \frac{C_{\text{iter}}}{N_{\text{iter}}} \tag{13}$$

where $C_{\text{iter}}$ is the current iteration number.

During a middle third of optimization process, an algorithm simulates the behavior of careful stepping, when the bears strengthen the pedal scents that were marked earlier. The stage can be used to filter positive solutions that may have been received during the earlier exploration phase which is represented by Equation (14). The update is based on best and worst solution to make the movement based on that information.

$$p_{i,j,k}^{new} = p_{i,j,k}^{old} + F_k \cdot (p_{j,k}^{best} - L_k \cdot p_{j,k}^{worst}) \tag{14}$$

where $p_{j,k}^{best}$ and $p_{j,k}^{worst}$ represent the jth best and worst pedal scent marks among all bear groups during the kth iteration, respectively. The step factor $F_k$, which determines the magnitude of the update, is calculated based on the occurrence factor $\theta_k$ as Equation (15).

$$F_k = (\beta_{1,k} \cdot \theta_k) \tag{15}$$

where $\beta_{1,k}$ is any random number in the range [0,1] during kth iteration. The step length $L_k$ during the kth iteration determines how pedal marks are adjusted based on the best and worst solutions. It takes a value of 1 or 2, guiding each male bear to step forward or backward accordingly. It is calculated Equation (16).

$$L_k = round(1 + \beta_{2,k}) \tag{16}$$

where $\beta_{2,k}$ is any random number uniformly distributed in the range [0,1].

The behavior of twisting feet is used in the last third of iterations. Every male bear adds to already formed pedal marks by kicking its feet into already existing depressions, further rendering the scent marks to the group more substantial and visible. Previous marks selection involves the best solutions and the worst solutions. The similarity between the angular velocity of a twist and a rotation is that in Equation (17).

$$\omega_{i,k} = 2\pi \cdot \theta_k + \gamma_{i,k} \tag{17}$$

where $\omega_{i,k}$ is the ith angular velocity of twist during kth iteration. $\gamma_{i,k}$ is a random number uniformly distributed in [0,1]. Bears twist their feet only on marks that are closer to the best and farther from the worst solutions. This behavior is modeled as Equation (18).

$$\boldsymbol{p}_{i,j,k}^{new} = \boldsymbol{p}_{i,j,k}^{old} + \omega_{i,k} \cdot \left(\boldsymbol{p}_{j,k}^{best} - |\boldsymbol{p}_{i,j,k}^{old}|\right) - \omega_{i,k} \cdot \left(\boldsymbol{p}_{j,k}^{worst} - |\boldsymbol{p}_{i,j,k}^{old}|\right) \tag{18}$$

It is worth mentioning here that after this phase, the selected better group of bears take part in the next phase.

In a sniffing behavior, all members in the group find their way through scenting marks. Each bear will be doing this involving themselves with randomly chosen marks of their own group so that being able to communicate and coordinate. This is mathematical who compares two randomly selected candidate solutions and updates the position of the bear in that manner as Equation (19).

$$\boldsymbol{p}_{m,j,k}^{new} = \begin{cases} \boldsymbol{p}_{m,j,k}^{old} + \lambda_{i,k} \cdot \left(\boldsymbol{p}_{m,j,k}^{old} - \boldsymbol{p}_{i,j,k}^{old}\right) & \text{if } f\left(\boldsymbol{p}_{m,k}^{old}\right) < f\left(\boldsymbol{p}_{n,k}^{old}\right) \\ \boldsymbol{p}_{m,j,k}^{old} + \lambda_{i,k} \cdot \left(\boldsymbol{p}_{i,j,k}^{old} - \boldsymbol{p}_{m,j,k}^{old}\right) & \text{if } f\left(\boldsymbol{p}_{n,k}^{old}\right) < f\left(\boldsymbol{p}_{m,k}^{old}\right) \end{cases} \tag{19}$$

The whole procedure of updating of pedal marks is dictated by behavior dynamics of brown bears such as gait, careful stepping, twisting and sniffing. These strategies with biologic motivation allow BBOA to find a desirable balance between exploration and exploitation, which is why it is very appropriate in the application to solving complex optimization problems, like hyperparameter tuning in YOLOv10. These behaviors are determined mathematically in a form of set of equation (10–19) based on the initial BBOA model [39].

**3.3.4 Hybrid GWO-BBOA: a novel approach to hyperparameter optimization for YOLOv10.** This work proposes the Hybrid GWO-BBOA as a significative optimization method, based on GWO and BBOA techniques, that overcomes their drawbacks such as local minima, premature convergence and loss of population diversity, can be applied to hyperparameter optimization of YOLO v10.

Although they have been used extensively in hyperparameter tuning, traditional metaheuristic algorithms like Genetic Algorithms (GA) and Particle Swarm Optimization (PSO) have well-known drawbacks. Premature convergence to suboptimal solutions, inadequate diversity maintenance, and the challenge of striking a balance between global and local search capabilities are some of these, particularly when used for intricate object detection tasks in dynamic surveillance environments. Similarly, although GWO and BBOA have unique benefits, when applied separately, they also have drawbacks. While BBOA, in spite of its adaptive behaviors, can become computationally costly or unstable in highly dynamic problem domains, GWO may converge too quickly if the search space is not sufficiently explored.

The creation of a hybrid GWO-BBOA framework that capitalizes on both of their advantages while reducing their drawbacks through complementary integration is motivated by this. Simultaneously, even with the exceptional speed and accuracy of YOLO-based models—particularly YOLOv10—they still have trouble identifying small-scale or unclear fire incidents, especially when there is occlusion, noise, or inadequate lighting. Suboptimal hyperparameter settings that restrict model generalization are frequently blamed for these drawbacks. By fusing the advantages of both algorithms, the suggested hybrid GWO-BBOA optimization framework directly addresses these drawbacks. GWO offers a well-structured leadership structure that promotes convergence while maintaining the caliber of the solution. In the meantime, BBOA increases diversity and lowers the chance of premature convergence by introducing dynamic exploration capabilities through behavioral mechanisms modeled after brown bears. As far as we are aware, this is the first study to use a hybrid GWO-BBOA algorithm for YOLOv10 tuning in the particular setting of arson detection. A new adaptive optimization technique made possible by this integration enhances the model's ability to identify important, uncommon, and context-sensitive events like arson.

This hybrid algorithm is designed based upon the hierarchical leadership and strategic hunting behavior mentioned in GWO and the exploration-refinement strategy of BBOA. The proposed hybrid GWO-BBOA integrates directional control from GWO with the dynamic occurrence of behavioral epochs in BBOA for space exploration and exploitation for hyperparameters. The structure of the hybrid optimization process comprises three main phases, each contributing to efficient hyperparameter tuning:

1. Initial Phase (GWO Leadership): Algorithm development starts with GWO's population structure in which the alpha, beta, and delta wolves create the search direction towards the promising region in the solution space. This makes the first phase of search more systematic and by making use of encirclement and convergence of GWO.

2. Refinement Phase: When the convergence of the population is around the optimum solutions, the algorithm shifts to another behavior modeled based on BBOA. This phase early establishes the pedal scent marking and sniffing mechanisms that facilitates fine tuning of solutions. The pedal scent marking is the guarantee to diversity as it continuously concentrates on the area of the promising solutions, while the sniffing phase refines the obtained information to avoid local optimum enough.

3. Feedback Mechanism: Feedback to the leadership structure of GWO comes from the refined solution of the hybrid structure where a BBOA phase precedes the GWO algorithm. This integration and iteration guarantee enhancement and affords a broad world view with comparable local exploitation.

 

These three phases collectively define the hybrid GWO-BBOA optimization strategy, which enhances exploration, refinement, and feedback-driven adaptation for hyperparameter tuning. A visual representation of this process is provided in Fig 5 (flowchart) and Fig 6 (pseudocode).

### 3.4 Model evaluation metrics

In order to evaluate the performance of YOLOv10 for detecting Arson incidents, we will determine the most significant evaluation metrics used in this process.

1. Precision is the proportion of correctly predicted positive observations to all predicted positives refer with: Eq. (20) [40].

$$\text{Precision} = \frac{TP}{TP + FP} \tag{20}$$

Where TP (True Positives), FP (False Positives).

2. Recall is the proportion of all observations in the actual class that were accurately predicted to be positive refer with: Eq. (21) [40].

$$\text{Recall} = \frac{TP}{TP + FN} \tag{21}$$

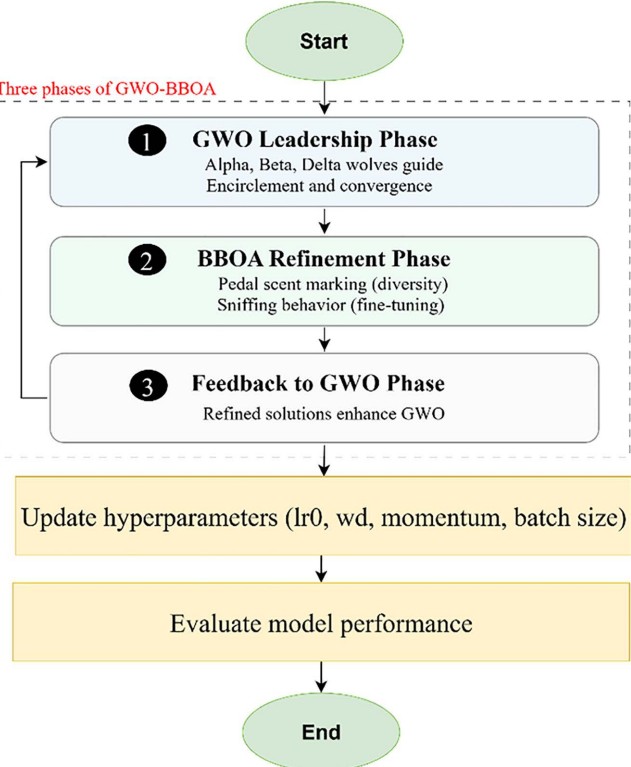

**Fig. 5. The Flowchart of the Hybrid GWO-BBOA Optimization Process.**

```
Input: Initial population of wolves and bears, search space boundaries, max iterations
Output: Optimal hyperparameters (lr0, weight decay, momentum)
Start:
Step 1: Initialize population
        1.1: For wolves: Use random initialization within boundaries.
        1.2: For bears: Use Eq. (10) and construct population using Eq. (11).
Step 2: Evaluate the fitness of each individual.
Step 3: Identify the alpha, beta, and delta wolves based on fitness.
Step 4: Repeat until convergence or maximum iterations:
    4.1: GWO Phase:
        4.1.1: Compute distances using Eq. (3):
        4.1.2: Update positions using Eq. (4):
    4.1.3: Compute coefficients using Eq. (5) and Eq. (6).
    4.1.4: Update each wolf's position w.r.t alpha, beta, delta using Eq. (7) -(9).
    4.2: BBOA Phase:
    4.2.1: Apply Pedal Scent Marking: Early iterations: Use Eq. (12) with
            occurrence factor Eq. (13)
    4.2.2: Conduct sniffing behavior to refine solutions, use Eq. (18), with twist
            velocity Eq. (17)
    4.2.3: Update positions of bears using Eq (19).
    4.2.4: Evaluate fitness of refined solutions.
    4.3: Feedback: Combine results from GWO and BBOA phases.
Step 5: Return the best solution (optimal hyperparameters).
End.
```

**Fig. 6. Pseudocode of the proposed Hybrid GWO-BBOA algorithm.**

Where TP (True Positives), FN (False Negatives).

3. Mean Average Precision (mAP) is the average precision score for each class. Average Precision (AP) is calculated for each class, and mAP is the mean of these values refer with: Eq. (22) [40].

$$mAP = \frac{1}{N}\sum_{i=1}^{N} AP_i \tag{22}$$

4. Mean Average Precision (mAP50-95) refers to the average mean Precision across Intersection over Union (IoU) thresholds ranging from 0.5 to 0.95 refer with: Eq. (23) [41].

$$mAP_{50-95} = \frac{1}{N}\sum_{t=0.5}^{0.95} AP_t \tag{23}$$

## 4. Results and discussion

The experimental results were obtained while executing the proposed method with the help of the Google Colab environment where some specifications are an A100 GPU with 40 GB RAM, 80 GB system RAM, and 200 GB disk space. Transfer learning was used to train the model, beginning with pre-trained weights on the COCO dataset which is quite well known to improve the performance of a model.

For the purpose of determining the optimal number of training epochs, experiments were conducted using 100 and 150 epochs. All other training parameters were kept at default values, including initial learning rate (lr0 = 0.02), training optimizer (AdamW), batch size [32], and weight decay (0.0005). According to the findings, training for 100 epochs provided

the most suitable balance between performance metrics (such as precision and recall) and training time (Table 2). Based on these results, a fixed value of 100 epochs was adopted for all subsequent experiments, including those involving hyperparameter variations. No early stopping strategy was applied, in order to ensure consistency and fair comparison across different settings.

To further support the experimental results, we provide a detailed plot (see Fig 7) showing the evolution of key training metrics and losses over the 100 training epochs. The plot demonstrates stable convergence in terms of classification, localization, and distribution-focused losses, as well as gradual improvement in evaluation metrics including mAP@0.5 and mAP@0.5:0.95.

### 4.1 YOLO v10 model size selection results

This section evaluates different YOLO v10 model sizes (nano (n), small(s), medium (m), big (b), large(l) and xlarge(x)) to determine which variant best serves the arson detection task. Starting with the YOLO v10n model, accuracy was moderate with a precision of 0.653, while the recall was 0.528. Despite its efficiency (0.339 hours), YOLO v10n is less effective for arson detection due to moderate recall and mAP performance. The small-sized YOLO v10s model outperformed the nano variant, achieving a precision of 0.727, recall of 0.470, mAP@0.50 of 0.558, and mAP@0.50:0.95 of 0.305, training time 0.377 hour. This makes it well-suited for arson detection scenarios making it a well-balanced choice between detection accuracy and computational efficiency. YOLO v10m and YOLO v10b provided similar performance but at the expense

**Table 2. Performance results of the YOLOv10s model on the Arson Detection Dataset.**

| epochs | Precision | Recall | mAP@0.50 | mAP@0.50:0.95 | Training time (hours) |
|---|---|---|---|---|---|
| Epoch = 100 | 0.727 | 0.47 | 0.558 | 0.305 | 0.377 |
| Epoch = 150 | 0.682 | 0.477 | 0.575 | 0.311 | 0.543 |

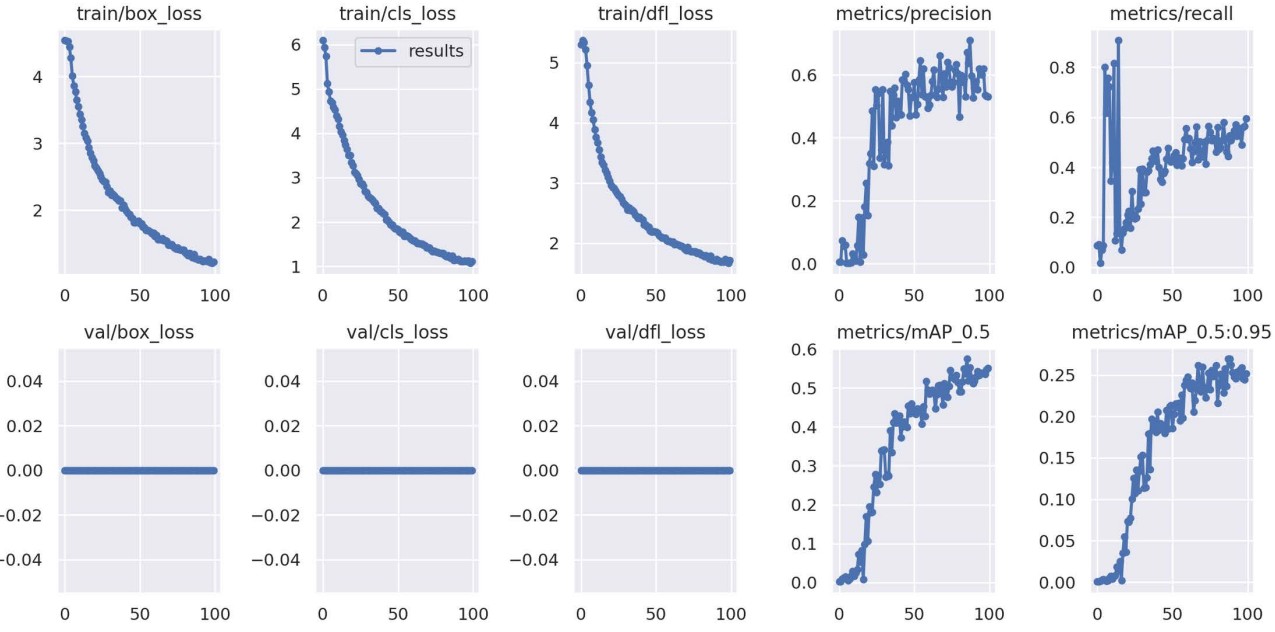

**Fig 7. Evolution of training losses and evaluation metrics (precision, recall, mAP@0.5, mAP@0.5:0.95) over 100 epochs.**

of increased training time. YOLO v10m achieved a precision of 0.662 and recall of 0.372 and the training time was 0.484 hours. Similarly, YOLO v10b showed balanced performance but required a longer training time, the network was trained for as long as 0.546 hours, making it less efficient. The largest model is YOLO v10x achieved only a slight improvement in precision (0.602) and recall (0.530) with the time of training 0.803 hours. This makes it less suitable for applications that require frequent feedback. Table 3 summarizes the performance results of all YOLO v10 versions evaluated on the arson detection dataset. According to the above analysis, conclude that hyperparameter optimization of the above models can be performed by selecting the YOLO v10s model to proceed further. Thus, it offers a satisfactory combination of accuracy and computation speed and therefore is well-adapted to the arson detection task.

## 4.2 Hyperparameter-tuning Results

The next step of this study entails hyperparameter optimization to determine the best setups of the specified problems. The tuning process leverages four optimization algorithms: PSO, GWO, BBOA, and the proposed hybrid GWO-BBOA. Table 4 below shows the list of hyperparameters and the choices that were considered for experimentation. Among such components, there are the lr0, lrf, mo, and wd. The ranges help in the selection of optimal sets of configurations; the optimization algorithms are used in the selection of the best set for the YOLOv10 model. In all experiments, the epochs were 20 and the population size used in optimization algorithms was 20 individuals to ensure that all comparative evaluations were consistent. Moreover, YOLO v10 was trained by AdamW optimizer with the batch size of 96.

Table 5 summarizes the performance results of YOLOv10s using different optimization algorithms, compared to the baseline configuration.

These results in Table 5 demonstrate that the hybrid GWO-BBOA algorithm has had better performance than the three other algorithms in most of the analyzing parameters. In particular, it scored the highest accuracy of 0.750, recall of 0.620, and mean Average Precision at 0.50 score of 0.705, evidence of its efficiency for enhancing the YOLOv10s model for detecting arson. The GWO algorithm followed closely, with strong performance across all metrics, particularly in mAP@0.50:0.95, thus getting the highest score of 0.403 from this particular site.

The performance of the YOLOv10 model with default (non-optimized) hyperparameters as a baseline to put these improvements in context. The baseline setup had a precision of 0.727, a recall of 0.470, and a mAP@0.50 of 0.558. When

**Table 3. Performance results of YOLO v10 versions on arson dataset.**

| model | Precision | Recall | F1-score | mAP@0.50 | mAP@0.50:0.95 | Training time (hours) |
|-------|-----------|--------|----------|----------|---------------|----------------------|
| YOLOv10n | 0.653 | 0.528 | 0.580 | 0.560 | 0.301 | 0.339 |
| YOLOv10s | 0.727 | 0.47 | 0.570 | 0.558 | 0.305 | 0.377 |
| YOLOv10m | 0.662 | 0.372 | 0.480 | 0.473 | 0.265 | 0.484 |
| YOLOv10b | 0.584 | 0.432 | 0.500 | 0.475 | 0.261 | 0.546 |
| YOLOv10l | 0.574 | 0.394 | 0.470 | 0.471 | 0.267 | 0.631 |
| YOLOv10x | 0.602 | 0.530 | 0.570 | 0.562 | 0.292 | 0.803 |

**Table 4. Parameters selected.**

| Parameter | Range |
|-----------|-------|
| lr0 | [0,1] |
| lrf | [0,1] |
| mo | [0,1] |
| wd | [0,1] |

**Table 5. Performance results of YOLOv10s using different optimization algorithms on the arson detection dataset.**

| algorithm | Precision | Recall | mAP@0.50 | mAP@0.50:0.95 |
|---|---|---|---|---|
| Baseline (YOLOv10 default hyperparameters) | 0.727 | 0.47 | 0.558 | 0.305 |
| PSO | 0.736 | 0.577 | 0.660 | 0.311 |
| GWO | 0.740 | 0.587 | 0.700 | 0.403 |
| BBOA | 0.666 | 0.610 | 0.682 | 0.394 |
| GWO_BBOA (proposed) | 0.750 | 0.620 | 0.705 | 0.400 |

you compare these numbers to the results of the GWO-BBOA method, you can see that the proposed optimization made both recall and detection accuracy much better.

There was a balanced performance of the BBOA algorithm, with a good strength in recall (0.610); this made it easily identify positive cases. Still, it performed somewhat lower in terms of precision and mAP coefficient averages compared to the observed results of the hybrid approach. Although PSO yielded high precision at 0.736, it performed fairly in recall and the mAP test.

The enhancement of the proposed hybrid GWO-BBOA can be attributed to its comprehensive exploration capacity of optimum solution compared to the GWO algorithm, complemented by the refinement potential of BBOA. This integration enabled taking a more efficient search for the optimal hyperparameter considering the trade-off between exploration and exploitation. All these results will be discussed further in the Result Discussion section.

## 4.3  Results discussion

The outputs of the hyperparameter tuning phase gave important data and understanding of the results of the various optimization algorithms used on the YOLOv10s model for arson detection. The analysis of the results, presented in Table 5, indicates clear differences in the performance metrics—precision, recall, mAP@0.50, and mAP@0.50:0.95–regarding all four optimization algorithms tested here: PSO, GWO, BBOA, and the analyzed hybrid GWO-BBOA. Before looking at how the different optimization algorithms affect performance, we should first look at how well the YOLOv10s model works with default (non-optimized) hyperparameters. The model got a precision of 0.727, a recall of 0.470, a mAP@0.50 of 0.558, and a mAP@0.50:0.95 of 0.305. The baseline precision is pretty high, but the low recall and mAP scores show that the model wasn't very good at finding all the cases of arson. These results show that optimization is necessary because improving recall and detection accuracy is very important for real-world uses like finding fires and arson. The next sections look at how each optimization algorithm dealt with these issues.

**4.3.1  Hybrid GWO-BBOA (proposed algorithm).**  The Hybrid GWO-BBOA algorithm demonstrated the best overall performance, achieving a precision of 0.750, recall of 0.620, mAP@0.50 of 0.705, and mAP@0.50:0.95 of 0.400. Enhancements in the performance can be owed to the fact that the combined algorithm is capable of both the global search and the local search. The GWO component deals with the crucial task of the proper coordination of the search process toward the direction of leading areas, and the BBOA component refines them. The technique of mixture of pedal scent marking and sniffing behavior from BBOA contributes toward offering the algorithm the capability of not ratcheting toward local minimizers by keeping the solution space sufficiently diversified. Similarly, the hierarchical nature of GWO keeps the population of solutions concentrated on the most promising areas due to the selection of the leaders.

A visual comparison between the ground truth annotations and the predicted bounding boxes for an arson scene is shown in Fig 8 to further demonstrate the detection performance of the suggested model. The image supports the high evaluation metrics reported by demonstrating how closely the optimized YOLOv10 using GWO-BBOA matches the actual locations of arson-related activity.

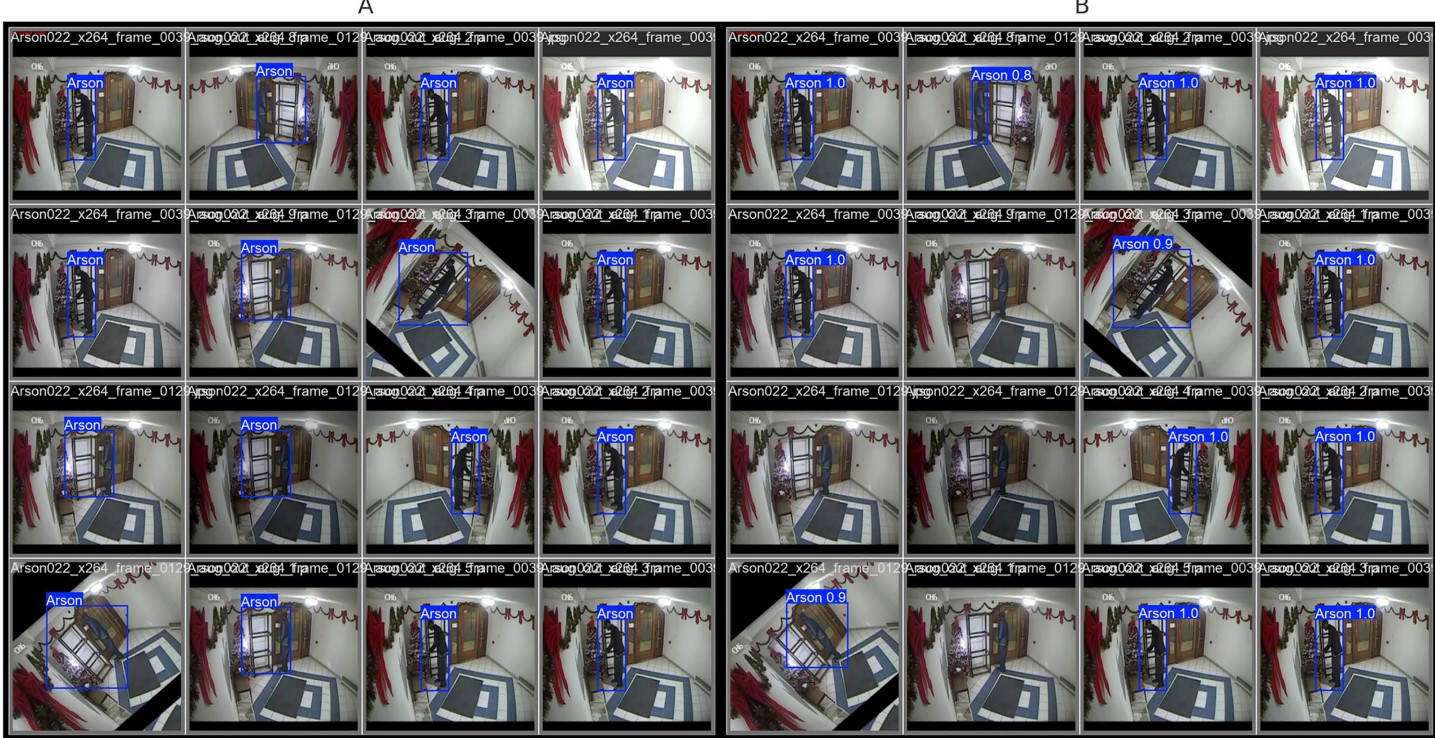

**Fig 8. Visual comparison of YOLOv10 arson detection results.** (A) Ground truth annotations for fire/arson events. (B) Predicted bounding boxes generated by YOLOv10 optimized with the proposed GWO-BBOA algorithm.

**4.3.2 Grey wolf optimizer.** The GWO algorithm also demonstrated strong performance, with a precision of 0.740, recall of 0.587, mAP@0.50 of 0.700, and the highest mAP@0.50:0.95 score of 0.403. This high performance owes to GWO's power hierarchy and leadership roles because it is regulated by the alpha, beta, and delta wolves in the search process. The encirclement and convergence behaviors enhance the algorithms for making the focused reduction of the optimal solution possible. Nonetheless, GWO can sometimes face low diversification in the solution space and hence get stuck at relatively slower rates for large, complex problems. Nevertheless, it has a relatively well-developed exploration capacity to maintain an accurate identification of arson events.

**4.3.3 Brown Bear Optimization Algorithm.** The results depicted that the BBOA algorithm had moderate performance throughout the parameters, with remarkable effectiveness in recall that is equal to 0.610, which also means that it is excellent in differentiating the positive cases of arson. Nonetheless, the proposed algorithm had slightly lower accuracy of 0.666 and mAP@0.50 of 0.682 as compared to the hybrid and GWO algorithms. The potential of BBOA is found in its rather peculiar phases, including the pedal scent marking and sniffing which enhance local exploration and make the solutions more refined. However, due to the lack of a more structured global search strategy as in GWO, the search operation's overall performance might be hamstrung in the pursuit of the best measure of precision and recall.

**4.3.4 Particle swarm optimization.** The high precision, marked as 0.736 in the present study, signified how the PSO algorithm successfully located the majority of the true positive cases. However, its recall of 0.577 and mAP@0.50:0.Out of total of 0.311, only 95 were relatively less than these. These findings on the performance of PSO can be attributed to the fact that particles are inclined to move towards the discovered best personal positions, as well as the best global solution

of the swarm. Despite this, it can spuriously make very fast progress and get stuck in local optimum which is especially detrimental in high dimensional and complex search space optimization.

**4.3.5 Statistical significance analysis.** In order to determine the statistical significance of the performance improvement obtained by the proposed Hybrid GWO-BBOA over individual optimization algorithms (PSO, GWO, and BBOA), five independent runs were performed of each algorithm. To compare the performance of the algorithms in all the major evaluation metrics, Precision, Recall, mAP@0.50, and mAP@0.50:0.95, the pairwise T-tests with Bonferroni correction were used. The findings, summarized in Table 6, indicate that the proposed GWO-BBOA is considerably better than PSO and BBOA in most metrics with highly significant differences in Precision, Recall and mAP@0.50. Conversely, the performance gap between GWO and GWO-BBOA is usually minor and insignificant in certain measures, including mAP@0.50 and mAP@0.50:0.95.

On the whole, these results prove that the hybrid optimization method is consistently better than PSO and BBOA and that it can perform as well as GWO in some settings. This shows the usefulness of hybrid optimization methods in improving the performance of deep learning models in detecting arson.

**4.3.6 Comparison with previous studies.** In order to test the capability of the proposed GWO-BBOA hybrid optimization approach for hyperparameter tuning in the YOLOv10 deep learning model for anomaly detection, it is compared with the works in the literature, which used other deep learning models for anomaly detection. While direct comparisons are challenging due to differences in datasets, models, and evaluation metrics, a relative performance analysis provides valuable insights into the advantages of our approach.

One of the most applicable studies in the recent past is that by Abbod et al. (2025) that used YOLOv9-s to detect arson events on surveillance video frames as shown in Table 7. Although their method proved that the YOLO-based models can be used in this area, they did not optimize default parameters. On a dataset of 2,182 images extracted in 53 videos, their model reached a precision of 0.445, recall of 0.507 and mAP@0.50 of 0.456. By contrast, we used GWO-BBOA metaheuristic tuning with YOLOv10 and achieved much better performance (precision = 0.750, recall = 0.620, mAP@0.50 = 0.705), which shows the obvious benefit of using more advanced methods of hyperparameter tuning.

While Singh et al. (2020), used the InceptionV3 + RNN model which is used to predict video sequences over time, our model focuses on real-time object detection within individual frames. Our interpretation of this distinction suggests that even though YOLOv10 is a very efficient tool in the task of detecting arson in still images, a sequential model, such as RNN, may be more effective in making inferences based on patterns at different frames in long term anomaly detection. The variation in approach to this indicates the necessity of establishing such models in accordance with the particular

**Table 6. Pairwise T-test results comparing the proposed GWO-BBOA with PSO, GWO, and BBOA.**

| Metric | Algorithm | T-value | p-value | Statistical Significance |
|---|---|---|---|---|
| Precision | PSO | −6.10 | 0.001 | Highly Significant |
| | GWO | −2.12 | 0.078 | Not Significant |
| | BBOA | −7.35 | <0.001 | Highly Significant |
| Recall | PSO | −5.48 | 0.002 | Highly Significant |
| | GWO | −3.34 | 0.024 | Significant |
| | BBOA | −2.98 | 0.033 | Significant |
| mAP@0.50 | PSO | −6.15 | 0.001 | Highly Significant |
| | GWO | −1.20 | 0.286 | Not Significant |
| | BBOA | −3.52 | 0.019 | Significant |
| mAP@0.50:0.95 | PSO | −4.65 | 0.006 | Significant |
| | GWO | 0.41 | 0.699 | Not Significant |
| | BBOA | −0.92 | 0.397 | Not Significant |

**Table 7. Presents a comparison between our study, Abbod et al. and Singh et al.**

| Criteria | Our Study | Abbod et al. [17] | Singh et al. [12] |
|---|---|---|---|
| Model | YOLOv10s + Optimization | YOLOv9-s | InceptionV3 + RNN |
| Optimization Techniques | PSO, GWO, BBOA, GWO-BBOA | Default hyperparameter tuning | Manual hyperparameter tuning |
| Application | Arson detection | Arson detection | General anomaly detection in surveillance videos |
| Data Type | Image frames (object-based detection) | Image frames from videos | Video sequences (temporal anomaly detection) |
| Dataset Size | 2,182 images from 53 videos | 2,182 images from 53 videos | Not specified |
| Best Performance | Precision = 0.750, Recall = 0.620, mAP@0.50 = 0.705 | Precision = 0.445, Recall = 0.507, mAP@0.50 = 0.456 | Precision, recall, and mAP@0.50 are not mentioned but an overall improvement Overall anomaly classification |
| Limitations | Focuses only on arson detection | Limited recall and precision; moderate performance under confidence tradeoff | No optimization of hyperparameters |

application demands, as real-time detection prospers from the object-based models like YOLO, while temporal models like RNN might be utilized for event-based anomaly detection.

These comparative findings open the door to a future study of hybrid models and general anomaly detection tasks, which are discussed in the following sections.

**4.3.7 Grad-CAM heatmaps for arson detection.** To assess the interpretability and effectiveness of the model, visual explanation techniques such as heatmaps are employed. These tools help clarify and validate the model's predictions by showing which areas of the image the model is focused on while making predictions. A heatmap transforms complex data into a vibrant, color-coded matrix [42]. As shown in Fig 9A, the original image is presented without any heatmap. In Fig 9B, we show bounding-box tightly enclosing both the person and the jug—to indicate the model's detected region of interest. Fig 9C, illustrates the heatmap generated by the YOLO v10 model, where attention is more concentrated on specific regions of the image, particularly the area around the fire. In the context of arson detection, the YOLO v10 model's decision-making process can be analyzed by interpreting its attention through these heatmaps. As fire detection requires distinguishing between accidental fires and deliberate fires (arson), the heatmaps serve to show which features such as human behavior, fire spread patterns, or unusual objects near the fire are deemed most relevant by the model. The red and yellow regions in the heatmap correlate with the features that the model associates with arson, confirming that it is focusing on the critical elements of the image that indicate intentional fire-setting behavior. Visualization tools like

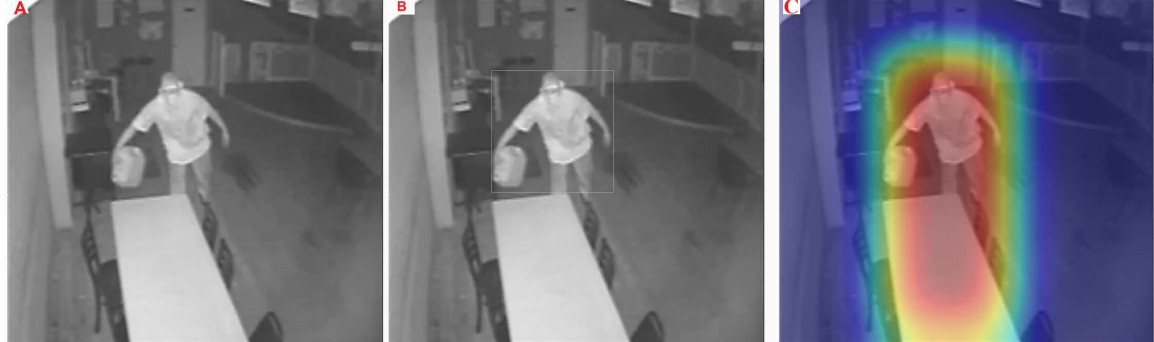

**Fig. 9. Visualized results for a single arson scene, showing (A) the original image, (B) YOLO v10 bounding-box, and (C) Grad-CAM heatmap overlay.**

heatmaps allow researchers and practitioners to understand the underlying reasons for a model's decisions, increasing confidence in its ability to detect arson and preventing FPs. However, with these interpretability techniques, we can assess the model's performance in different scenarios. Depending on the situation the model should concentrate more on human interactions and behaviors and less on suspicious activities in case of purposeful fire and vice versa in case of an accidental fire. This flexibility in the model's interpretability is crucial for ensuring accurate and reliable arson detection in real-world environments.

Finally, the experimental evaluation of the YOLO v10 model showed that it is capable of performing effective arson detection when appropriately sized and using advanced hyperparameter tuning techniques.

## 5.Conclusions and suggestions for future work

As for the contribution of this research, it focuses on the performance analysis of the YOLOv10s model for arson detection, considering the improvement of hyperparameter tuning techniques and optimization algorithms. Through a comprehensive experimental setup, several key findings emerged:

- As compared to all other optimization techniques, hybrid GWO-BBOA was found to be the finest in precision (0.750), recall (0.620), and mAP@0.50 (0.705) to optimize all parameters and to have an optimum balance between exploration and exploitation.

- The GWO algorithm followed closely, particularly excelling in mAP@0.50:0.95 (0.403) and provided evidence of its ability to deliver complex search spaces albeit at the costs of occasional lack of diversity.

- The cases detected aggressively and accurately by using both the pedal scent marking and sniffing proved that the BBOA algorithm has high recall which was measured 0.610. However, it did not receive the global direction that would result into fairly balanced performance.

- For the PSO algorithm, recall, and mAP were compromised while precision obtained a value of 0.736, mainly because of a problem attributed to the algorithm that is early convergence to suboptimal solutions.

These results show is that there a need to use metaheuristic algorithms with both global and local search techniques, as is the case with GWO-BBOA for enhancing arson detection in real time analysis and computational performance. Although the model has demonstrated strong performance in controlled environments, deploying YOLOv10 with the GWO-BBOA optimized parameters in real-world systems introduces practical challenges. One major concern is latency, especially in time-sensitive scenarios such as fire or arson detection. The model must deliver inference within milliseconds to be viable for live surveillance systems. YOLOv10s, being a lightweight variant, offers promising latency levels (sub-100ms on modern GPUs), but edge deployment requires further compression and quantization. Moreover, deploying the model on edge devices like security cameras or IoT systems brings resource constraints such as limited memory, processing power, and battery life. While YOLOv10s is portable, real-time performance on edge may require pruning or using accelerators like NVIDIA Jetson or Google Coral. Additional challenges include ensuring robustness across varied environmental conditions, maintaining data privacy, and integrating seamlessly with legacy surveillance infrastructure.

Additionally, future research can focus on the following directions:

1. obtaining a far more extensive and diverse collection of fire video footage, ideally encompassing a range of indoor and outdoor environments, camera perspectives, and sources of ignition. Simultaneously, we advise investigating sophisticated synthetic data generation methods, like generative adversarial networks (GANs), to produce lifelike fire simulations that enhance actual video. To improve model generalization and guarantee reliable performance in arbitrary surveillance scenarios, such efforts will be essential.

2. performing cross-validation and testing the model on external datasets to improve generalizability and ensure that the results are reproducible across different data distributions.

3. Adaptive Optimization: Exploring variations of the proposed GWO-BBOA learning algorithm as adaptive or self-tuning versions that would allow the proper tuning of exploration and exploitation throughout the training process.

4. Integration with Real-Time Systems: it is possible to consider how the presented model can be connected to real fire detection systems, analyze its effectiveness in the conditions of real usage, such as using it in surveillance cameras in different industrial and living zones.

5. Multi-Event Detection: Expanding the model to work for different types of events including explosions or smoke would further enhance the model's use. It was possible to refine hyperparameters in multi-class object detection by applying the changes in hybrid optimization.

## Author contributions

**Project administration:** Matheel E. Abdulmunim.

**Software:** Ali Abbas Abbod.

**Supervision:** Matheel E. Abdulmunim.

**Validation:** Matheel E. Abdulmunim.

**Writing – original draft:** Ali Abbas Abbod.

**Writing – review & editing:** Ismail A Mageed.

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
