## [Decision Letter · Decision Letter 0]

8 Jul 2025

Dear Dr. Abbod,

Please note that, at this level of publication, we have see that, one or to reviewers have asked for particular citations, this is not mandatory for the authors, it is the responsability of the authors to check if the forced references are relevant to the study or not. the authors are not forced to report any citation that is not relevant to the study and a detail reply with solid justification should be provided.

We look forward to receiving your revised manuscript.

Kind regards,

Salim Heddam

Academic Editor

PLOS ONE

A clean copy of the edited manuscript (uploaded as the new *manuscript* file)”.

6.  Please include a copy of Table 5 which you refer to in your text on page 23.

Comments from the Editorial Office:

One or more of the reviewers has recommended that you cite specific previously published works. Members of the editorial team have determined that the works referenced are not directly related to the submitted manuscript. As such, please note that it is not necessary or expected to cite the works requested by the reviewer. 

Furthermore, please do not respond to Reviewer 1 as we have concerns that have utilised AI tools in their review.

Reviewer 1#:

This manuscript presents an innovative approach for optimizing hyperparameters of the YOLOv10 deep learning model for arson detection. The authors propose a hybrid optimization algorithm that combines the Grey Wolf Optimization (GWO) and Brown Bear Optimization Algorithm (BBOA), referred to as GWO-BBOA. This hybrid algorithm aims to improve the model's precision and recall in detecting arson events by fine-tuning hyperparameters such as learning rate, momentum, and weight decay. The method was evaluated on a custom arson detection dataset of 2,182 images. The results indicate that GWO-BBOA outperforms traditional optimization methods such as PSO, GWO, and BBOA, showing better performance in precision, recall, and mAP (mean Average Precision).

Significance of the Study:

The study highlights the growing importance of real-time anomaly detection systems, particularly for safety-critical applications such as arson detection. By optimizing YOLOv10 using a hybrid optimization approach, the manuscript demonstrates the potential of advanced machine learning and optimization techniques in improving model performance for real-time applications. The contribution is significant, as it addresses the challenge of balancing high precision and low false-positive rates in a domain where false alarms can have serious consequences.

Content and Structure Review

Abstract:

The abstract effectively summarizes the key contributions of the study. It clearly outlines the methodology and provides quantitative results (precision, recall, and mAP), which is essential for understanding the model's performance. However, it could be more concise in stating the results and their implications. Furthermore, it would be helpful to briefly mention the limitations of the current model, such as false positives and scalability issues, which are addressed in the manuscript.

Introduction:

The introduction provides a strong rationale for the study, clearly stating the importance of accurate arson detection in high-risk areas. However, the manuscript would benefit from a more detailed comparison with existing methods and an explanation of why YOLOv10 was selected as the model of choice over other object detection algorithms. Additionally, references to recent works on hybrid optimization algorithms, such as DOI: 10.54216/MOR.030205, would strengthen the argument for the effectiveness of the hybrid approach.

Conclusion:

The conclusion summarizes the key findings well. However, it could provide more insight into future work. For example, the authors could discuss the potential for integrating this model with real-time fire detection systems, expanding the dataset, or improving the model's performance under varying environmental conditions.

Literature Review and Citation Updates

Literature Review:

The literature review provides a comprehensive background on arson detection and optimization algorithms used in deep learning models. It discusses the limitations of previous studies and justifies the need for a hybrid optimization approach. However, it would benefit from the inclusion of more recent works on hybrid algorithms in deep learning, such as DOI: 10.54216/JAIM.090102, which could provide further context to the hybrid methodology employed in this study.

Citations:

The manuscript includes a solid list of references, but it could benefit from the inclusion of more recent studies on hybrid machine learning approaches, such as DOI: 10.54216/MOR.030205, to provide a broader perspective on the evolution of optimization techniques in deep learning models.

https://doi.org/10.1016/j.eswa.2023.122147

https://doi.org/10.54216/JAIM.090102

https://doi.org/10.54216/MOR.030205

https://doi.org/10.1007/s11540-024-09717-0

https://doi.org/10.32604/cmc.2023.031723

Technical Review

Methodology and Algorithms:

The methodology is well-described, with a clear explanation of the proposed hybrid optimization algorithm, GWO-BBOA. However, more detailed explanations of the individual algorithms (GWO, BBOA, and their hybridization) would improve the clarity of the methodology. For example, a more thorough explanation of the mechanism behind the pedal scent marking and sniffing behaviors in BBOA would help readers understand why these steps are important in the optimization process.

Hyperparameter Tuning and Validation:

The manuscript briefly mentions hyperparameter tuning but lacks sufficient detail about the exact search space used for each parameter. Including more details about how the hyperparameters were tuned and how the results were validated (e.g., through cross-validation) would improve the rigor of the study. The authors could also discuss the computational cost of training with these optimization techniques.

Performance Evaluation

Result Presentation:

The results section provides a clear presentation of the performance metrics for the proposed model. However, additional performance metrics such as confusion matrices and ROC curves would help assess the model's classification performance in more detail. Visual comparisons of the proposed GWO-BBOA method with other algorithms would also enhance the reader’s understanding of the model's strengths and weaknesses.

Visualizations:

The manuscript includes some useful tables and figures, but additional visualizations comparing the results of different algorithms would improve the presentation. A graphical representation of precision-recall curves or a heatmap of mAP scores would be particularly helpful.

Reviewer 2#:

The paper presents a promising hybrid optimization approach for arson detection using YOLOv10. However, it requires substantial revision to improve clarity, methodological transparency, and critical discussion of limitations and applicability. Addressing these comments will significantly enhance the quality and impact of the manuscript.

Overall Strengths:

The paper tackles an important real-world problem (arson detection) using recent advancements in deep learning and optimization.

It is well-structured and mostly adheres to academic conventions.

The hybrid GWO-BBOA algorithm is a novel contribution with demonstrated performance gains.

Extensive experiments and comparisons are made, including model variants and multiple optimization strategies.

Comments for Improvement:

Clarity and English Language Quality

The manuscript contains grammatical and syntactic issues throughout. Many sentences are long, redundant, or awkwardly phrased. A professional English language editing service is highly recommended.

Example: “Besides, damage early minimizes loss of lives, obedience to authorities…” → This is unclear. Likely intended to be something like: “Early damage detection minimizes loss of life and ensures compliance with safety regulations.”

Suggestion - Please send for proofread.

Methodological Justification

The choice of hyperparameters (e.g., lr0, lrf, mo, wd) should be better justified based on domain knowledge or ablation studies.

The rationale behind specific values (e.g., population size = 20, epochs = 20) should be clarified—were these empirically determined?

Novelty and Comparison

While the hybrid GWO-BBOA shows better results than individual algorithms, it’s unclear how statistically significant these differences are. A statistical significance test (e.g., t-test or ANOVA) should be included.

The comparison with Singh et al. (2020) is weak, as the models, datasets, and objectives differ greatly. Instead, a comparison with similar object detection or fire/arson detection papers using YOLO variants would strengthen the discussion.

Lack of Visual Results

There is a notable absence of qualitative visual results (e.g., bounding box detection outputs for arson scenes). Including detection outputs for different scenarios (e.g., low light, occlusions) would demonstrate practical robustness.

Dataset Limitations and Augmentation

Although augmentation expands the dataset, the original number of 290 frames from 53 videos is still quite small for training deep models. This limitation needs to be acknowledged and discussed more thoroughly, especially its effect on model generalization.

Real-World Deployment Discussion

The conclusion briefly mentions integration with real-time systems. However, more elaboration is needed: What are the latency constraints? Can the model run on edge devices (e.g., security cameras)? What challenges are foreseen in such deployments?

Minor Comments and Technical Corrections:

Abstract: Avoid listing too many metrics in the abstract unless absolutely necessary.

Introduction: The statement "Accidental fires can also cause disastrous effects such causing fires on purpose" is unclear and contradictory.

Section 2.3.4: The pseudocode is helpful but could benefit from line numbers and a figure/table format for clarity.

References: Make sure the reference style is consistent. Some citations are numbered inline; others are formatted inconsistently.

Reviewer 3#:

While results are convincing, cross-validation or testing on a secondary dataset would enhance the generalizability of the claims.

Including confidence intervals or statistical significance tests could further strengthen the validity of the comparisons.

Single-run evaluation is implied (no mention of repeated trials or random seeds), which can limit the reproducibility and reliability of the reported performance.

Reviewer 4#:

Strengths:

1. The paper addresses an important and practical problem: arson detection in real-time surveillance settings using deep learning techniques.

2. The proposed hybrid optimization method (GWO-BBOA) applied to YOLOv10 is interesting, and experimental results are reported to support its effectiveness.

Major Concerns and Suggestions for Improvement:

1. Lack of Related Work Section:

The manuscript does not include a dedicated Related Work section. A detailed comparison with prior work—especially in the areas of fire/arson detection and other optimization methods—is crucial for positioning the novelty and contribution of this work.

2. Insufficient Discussion of Novelty and Challenges:

The introduction briefly states the use of YOLOv10 and the hybrid optimization method, but it does not clearly articulate the technical challenges being addressed that prior works have not solved. Please elaborate on:

a) What specific limitations in previous models or optimization strategies this work aims to overcome?

b) Why GWO-BBOA is particularly suitable in this context?

3. No Visualization of Detection Results:

To assess the practical effectiveness of the proposed method, the paper should include visual examples of detection results (e.g., bounding boxes overlaid on frames with fire/arson events). This helps reviewers and readers evaluate qualitative performance.

4. Lack of Model Input/Output Description:

The paper does not clearly explain what kind of data is used as input to the model (e.g., image resolution, frame rate, RGB vs. thermal, etc.) and what format the model outputs. Please provide a clearer overview of the pipeline, including preprocessing (if any) and output interpretation.

5. No Code Availability:

To ensure reproducibility and to support future research, the authors should provide a public link to their code. This is particularly important for works involving novel optimization strategies applied to established models.

Reviewer 5#:

The manuscript explores hyperparameter optimization strategies to enhance the performance of YOLOv10, which is a lightweight yet fast real-time object detection model. The authors proposed a new method, GWO-BBOA, that integrates Grey Wolf Optimization (GWO) and the Brown Bear Optimization Algorithm (BBOA) to effectively tune the model's hyperparameters.

Comments:

The reproduction of Figure 2 from the original YOLOv10 paper is not presented clearly. All four figures are in low resolution, which affects readability. It is strongly recommended to use high-resolution or vector graphics when applicable.

Baselines are essential for evaluation. Please include the training results of YOLOv10 using the original hyperparameters in Table 3 for comparison.

It seems that two pictures augmented from the same original picture were split across the training set and test sets. Arson002_x264_frame_0060.jpg from the test set and Arson002_x264_frame_0059_aug_out_aug_1.png from the training set are very similar. To prevent data leakage and ensure the validity of results, such overlap should be avoided.

In line 339, it’s unclear whether 100 epochs were applied to all training. Since different hyperparameters may lead to the model convergence at different epochs. It’s advisable to report the stopping strategy used in this research. Reporting metrics such as mAP or loss at each epoch would support the experimental results.

Regarding Table 4, more relevant and recent works may serve as better points of comparison. Suggested alternatives include: https://doi.org/10.1016/j.eswa.2023.119741 and https://doi.org/10.3390/app14135841.

The program mentioned on line 138 needs to be properly cited.

Grammatical correctness is needed at lines 35, 149, and 333.

A range should be given at each line of 389, 390, and 392.

Reviewers' comments:

Reviewer's Responses to Questions

**Comments to the Author**

1. Is the manuscript technically sound, and do the data support the conclusions?

Reviewer #1: Yes

Reviewer #2: Partly

Reviewer #3: Yes

Reviewer #4: Yes

Reviewer #5: Partly

2. Has the statistical analysis been performed appropriately and rigorously?

Reviewer #1: Yes

Reviewer #2: Yes

Reviewer #3: Yes

Reviewer #4: Yes

Reviewer #5: N/A

3. Have the authors made all data underlying the findings in their manuscript fully available?

Reviewer #1: Yes

Reviewer #2: Yes

Reviewer #3: Yes

Reviewer #4: Yes

Reviewer #5: Yes

4. Is the manuscript presented in an intelligible fashion and written in standard English?

Reviewer #1: Yes

Reviewer #2: No

Reviewer #3: Yes

Reviewer #4: Yes

Reviewer #5: Yes

Reviewer #1: This manuscript presents an innovative approach for optimizing hyperparameters of the YOLOv10 deep learning model for arson detection. The authors propose a hybrid optimization algorithm that combines the Grey Wolf Optimization (GWO) and Brown Bear Optimization Algorithm (BBOA), referred to as GWO-BBOA. This hybrid algorithm aims to improve the model's precision and recall in detecting arson events by fine-tuning hyperparameters such as learning rate, momentum, and weight decay. The method was evaluated on a custom arson detection dataset of 2,182 images. The results indicate that GWO-BBOA outperforms traditional optimization methods such as PSO, GWO, and BBOA, showing better performance in precision, recall, and mAP (mean Average Precision).

Significance of the Study:

The study highlights the growing importance of real-time anomaly detection systems, particularly for safety-critical applications such as arson detection. By optimizing YOLOv10 using a hybrid optimization approach, the manuscript demonstrates the potential of advanced machine learning and optimization techniques in improving model performance for real-time applications. The contribution is significant, as it addresses the challenge of balancing high precision and low false-positive rates in a domain where false alarms can have serious consequences.

Content and Structure Review

Abstract:

The abstract effectively summarizes the key contributions of the study. It clearly outlines the methodology and provides quantitative results (precision, recall, and mAP), which is essential for understanding the model's performance. However, it could be more concise in stating the results and their implications. Furthermore, it would be helpful to briefly mention the limitations of the current model, such as false positives and scalability issues, which are addressed in the manuscript.

Introduction:

The introduction provides a strong rationale for the study, clearly stating the importance of accurate arson detection in high-risk areas. However, the manuscript would benefit from a more detailed comparison with existing methods and an explanation of why YOLOv10 was selected as the model of choice over other object detection algorithms. Additionally, references to recent works on hybrid optimization algorithms, such as DOI: 10.54216/MOR.030205, would strengthen the argument for the effectiveness of the hybrid approach.

Conclusion:

The conclusion summarizes the key findings well. However, it could provide more insight into future work. For example, the authors could discuss the potential for integrating this model with real-time fire detection systems, expanding the dataset, or improving the model's performance under varying environmental conditions.

Literature Review and Citation Updates

Literature Review:

The literature review provides a comprehensive background on arson detection and optimization algorithms used in deep learning models. It discusses the limitations of previous studies and justifies the need for a hybrid optimization approach. However, it would benefit from the inclusion of more recent works on hybrid algorithms in deep learning, such as DOI: 10.54216/JAIM.090102, which could provide further context to the hybrid methodology employed in this study.

Citations:

The manuscript includes a solid list of references, but it could benefit from the inclusion of more recent studies on hybrid machine learning approaches, such as DOI: 10.54216/MOR.030205, to provide a broader perspective on the evolution of optimization techniques in deep learning models.

https://doi.org/10.1016/j.eswa.2023.122147

https://doi.org/10.54216/JAIM.090102

https://doi.org/10.54216/MOR.030205

https://doi.org/10.1007/s11540-024-09717-0

https://doi.org/10.32604/cmc.2023.031723

Technical Review

Methodology and Algorithms:

The methodology is well-described, with a clear explanation of the proposed hybrid optimization algorithm, GWO-BBOA. However, more detailed explanations of the individual algorithms (GWO, BBOA, and their hybridization) would improve the clarity of the methodology. For example, a more thorough explanation of the mechanism behind the pedal scent marking and sniffing behaviors in BBOA would help readers understand why these steps are important in the optimization process.

Hyperparameter Tuning and Validation:

The manuscript briefly mentions hyperparameter tuning but lacks sufficient detail about the exact search space used for each parameter. Including more details about how the hyperparameters were tuned and how the results were validated (e.g., through cross-validation) would improve the rigor of the study. The authors could also discuss the computational cost of training with these optimization techniques.

Performance Evaluation

Result Presentation:

The results section provides a clear presentation of the performance metrics for the proposed model. However, additional performance metrics such as confusion matrices and ROC curves would help assess the model's classification performance in more detail. Visual comparisons of the proposed GWO-BBOA method with other algorithms would also enhance the reader’s understanding of the model's strengths and weaknesses.

Visualizations:

The manuscript includes some useful tables and figures, but additional visualizations comparing the results of different algorithms would improve the presentation. A graphical representation of precision-recall curves or a heatmap of mAP scores would be particularly helpful.

Reviewer #2: The paper presents a promising hybrid optimization approach for arson detection using YOLOv10. However, it requires substantial revision to improve clarity, methodological transparency, and critical discussion of limitations and applicability. Addressing these comments will significantly enhance the quality and impact of the manuscript.

Overall Strengths:

The paper tackles an important real-world problem (arson detection) using recent advancements in deep learning and optimization.

It is well-structured and mostly adheres to academic conventions.

The hybrid GWO-BBOA algorithm is a novel contribution with demonstrated performance gains.

Extensive experiments and comparisons are made, including model variants and multiple optimization strategies.

Comments for Improvement:

Clarity and English Language Quality

The manuscript contains grammatical and syntactic issues throughout. Many sentences are long, redundant, or awkwardly phrased. A professional English language editing service is highly recommended.

Example: “Besides, damage early minimizes loss of lives, obedience to authorities…” → This is unclear. Likely intended to be something like: “Early damage detection minimizes loss of life and ensures compliance with safety regulations.”

Suggestion - Please send for proofread.

Methodological Justification

The choice of hyperparameters (e.g., lr0, lrf, mo, wd) should be better justified based on domain knowledge or ablation studies.

The rationale behind specific values (e.g., population size = 20, epochs = 20) should be clarified—were these empirically determined?

Novelty and Comparison

While the hybrid GWO-BBOA shows better results than individual algorithms, it’s unclear how statistically significant these differences are. A statistical significance test (e.g., t-test or ANOVA) should be included.

The comparison with Singh et al. (2020) is weak, as the models, datasets, and objectives differ greatly. Instead, a comparison with similar object detection or fire/arson detection papers using YOLO variants would strengthen the discussion.

Lack of Visual Results

There is a notable absence of qualitative visual results (e.g., bounding box detection outputs for arson scenes). Including detection outputs for different scenarios (e.g., low light, occlusions) would demonstrate practical robustness.

Dataset Limitations and Augmentation

Although augmentation expands the dataset, the original number of 290 frames from 53 videos is still quite small for training deep models. This limitation needs to be acknowledged and discussed more thoroughly, especially its effect on model generalization.

Real-World Deployment Discussion

The conclusion briefly mentions integration with real-time systems. However, more elaboration is needed: What are the latency constraints? Can the model run on edge devices (e.g., security cameras)? What challenges are foreseen in such deployments?

Minor Comments and Technical Corrections:

Abstract: Avoid listing too many metrics in the abstract unless absolutely necessary.

Introduction: The statement "Accidental fires can also cause disastrous effects such causing fires on purpose" is unclear and contradictory.

Section 2.3.4: The pseudocode is helpful but could benefit from line numbers and a figure/table format for clarity.

References: Make sure the reference style is consistent. Some citations are numbered inline; others are formatted inconsistently.

Reviewer #3: While results are convincing, cross-validation or testing on a secondary dataset would enhance the generalizability of the claims.

Including confidence intervals or statistical significance tests could further strengthen the validity of the comparisons.

Single-run evaluation is implied (no mention of repeated trials or random seeds), which can limit the reproducibility and reliability of the reported performance.

Reviewer #4: Strengths:

1. The paper addresses an important and practical problem: arson detection in real-time surveillance settings using deep learning techniques.

2. The proposed hybrid optimization method (GWO-BBOA) applied to YOLOv10 is interesting, and experimental results are reported to support its effectiveness.

Major Concerns and Suggestions for Improvement:

1. Lack of Related Work Section:

The manuscript does not include a dedicated Related Work section. A detailed comparison with prior work—especially in the areas of fire/arson detection and other optimization methods—is crucial for positioning the novelty and contribution of this work.

2. Insufficient Discussion of Novelty and Challenges:

The introduction briefly states the use of YOLOv10 and the hybrid optimization method, but it does not clearly articulate the technical challenges being addressed that prior works have not solved. Please elaborate on:

a) What specific limitations in previous models or optimization strategies this work aims to overcome?

b) Why GWO-BBOA is particularly suitable in this context?

3. No Visualization of Detection Results:

To assess the practical effectiveness of the proposed method, the paper should include visual examples of detection results (e.g., bounding boxes overlaid on frames with fire/arson events). This helps reviewers and readers evaluate qualitative performance.

4. Lack of Model Input/Output Description:

The paper does not clearly explain what kind of data is used as input to the model (e.g., image resolution, frame rate, RGB vs. thermal, etc.) and what format the model outputs. Please provide a clearer overview of the pipeline, including preprocessing (if any) and output interpretation.

5. No Code Availability:

To ensure reproducibility and to support future research, the authors should provide a public link to their code. This is particularly important for works involving novel optimization strategies applied to established models.

Reviewer #5: The manuscript explores hyperparameter optimization strategies to enhance the performance of YOLOv10, which is a lightweight yet fast real-time object detection model. The authors proposed a new method, GWO-BBOA, that integrates Grey Wolf Optimization (GWO) and the Brown Bear Optimization Algorithm (BBOA) to effectively tune the model's hyperparameters.

Comments:

The reproduction of Figure 2 from the original YOLOv10 paper is not presented clearly. All four figures are in low resolution, which affects readability. It is strongly recommended to use high-resolution or vector graphics when applicable.

Baselines are essential for evaluation. Please include the training results of YOLOv10 using the original hyperparameters in Table 3 for comparison.

It seems that two pictures augmented from the same original picture were split across the training set and test sets. Arson002_x264_frame_0060.jpg from the test set and Arson002_x264_frame_0059_aug_out_aug_1.png from the training set are very similar. To prevent data leakage and ensure the validity of results, such overlap should be avoided.

In line 339, it’s unclear whether 100 epochs were applied to all training. Since different hyperparameters may lead to the model convergence at different epochs. It’s advisable to report the stopping strategy used in this research. Reporting metrics such as mAP or loss at each epoch would support the experimental results.

Regarding Table 4, more relevant and recent works may serve as better points of comparison. Suggested alternatives include: https://doi.org/10.1016/j.eswa.2023.119741 and https://doi.org/10.3390/app14135841.

The program mentioned on line 138 needs to be properly cited.

Grammatical correctness is needed at lines 35, 149, and 333.

A range should be given at each line of 389, 390, and 392.

**Do you want your identity to be public for this peer review?** For information about this choice, including consent withdrawal, please see our Privacy Policy

Reviewer #1: No

Reviewer #2: No

Reviewer #3: **Yes: ** V. V. Subrahmanyam

Reviewer #4: No

Reviewer #5: No

---

## [Author Response · Author response to Decision Letter 1]

11 Aug 2025

Response to Reviewers

Manuscript Title: Optimizing Hyperparameters of YOLOv10 for Arson Detection Using Advanced Optimization Algorithms

Manuscript Number: PONE-D- 25-07942

We would like to express our sincere appreciation to the editor and the reviewers for their insightful comments and valuable suggestions, which have greatly contributed to enhancing the clarity and overall quality of our manuscript. We have carefully addressed all concerns and implemented the recommended changes accordingly. The following sections present our detailed responses and highlight the modifications made based on the reviewers' feedback. We are grateful for your constructive input and the opportunity to improve our work through this review process.

Academic Editor

We acknowledge your comments and have taken them into sincere consideration. Below is our response to each of the points raised, marked using yellow color in the Revised Manuscript with Track Changes.

Please note that PLOS ONE has specific guidelines on code sharing for submissions in which author-generated code underpins the findings in the manuscript. In these cases, we expect all author-generated code to be made available without restrictions upon publication of the work. Please review our guidelines at https://journals.plos.org/plosone/s/materials-and-software-sharing#loc-sharing-code and ensure that your code is shared in a way that follows best practice and facilitates reproducibility and reuse.

Response: We confirm that all author-generated code has been made publicly available on GitHub in accordance with PLOS ONE’s code sharing policy:

https://github.com/AliAbbasAbbod/YOLOv10-Arson-Optimization

We suggest you thoroughly copyedit your manuscript for language usage, spelling, and grammar. If you do not know anyone who can help you do this, you may wish to consider employing a professional scientific editing service.

Response: The manuscript has been carefully proofread for grammar and clarity. All suggested language issues have been addressed and corrected in the revised version.

We note that your Data Availability Statement is currently as follows: [All relevant data are within the manuscript and its Supporting Information files.

Response: We confirm that the minimal dataset required to replicate our results has been made publicly available via the following Kaggle repository:

https://www.kaggle.com/datasets/aliabbasabbod/arson-dataset

Additionally, supporting files including training logs and evaluation metrics are included in the Supplementary Materials.

Comment: One or more of the reviewers has recommended that you cite specific previously published works. Members of the editorial team have determined that the works referenced are not directly related to the submitted manuscript. As such, please note that it is not necessary or expected to cite the works requested by the reviewer.

Response: We have considered the references suggested by the reviewers, but only incorporated those that are strongly relevant to the objectives and scope of this study.

Comment: Please include a copy of Table 5 which you refer to in your text on page 23.

Response: Thank you for your comment. We have ensured that all tables, including Table 5, are properly included and correctly referenced in the revised manuscript. 

Reviewer 2

We acknowledge your comments and have taken them into sincere consideration. Below is our response to each of the points raised, marked using yellow color in the revised manuscript with track changes.

• Comment: Clarity and English Language Quality: The manuscript contains grammatical and syntactic issues throughout. Many sentences are long, redundant, or awkwardly phrased. A professional English language editing service is highly recommended. Example: “Besides, damage early minimizes loss of lives, obedience to authorities…” → This is unclear. Likely intended to be something like: “Early damage detection minimizes loss of life and ensures compliance with safety regulations.”.

Response: We have carefully proofread and revised the entire manuscript, including the abstract, methodology, results, and other sections, to enhance grammar, syntax, and readability.

Revised part: Early detection of damages reduces loss of life, compliance with authorities and preservation of the environment.

• Comment: The choice of hyperparameters (e.g., lr0, lrf, mo, wd) should be better justified based on domain knowledge or ablation studies.

Response: The hyperparameters (lr0, lrf, mo, and wd) were selected due to their well-documented and significant impact on convergence speed, generalization, and overfitting control, as highlighted in recent deep learning literature. In the revised manuscript (Section 2.3, Table 2.3), we explicitly justified their importance and provided their recommended ranges based on previous studies. Moreover, these parameters were prioritized because of their critical role in real-time object detection tasks, such as arson detection, where rapid response and high precision are essential.

Revised part: In modern ML and DL sets, hyperparameters are important in determining model performance. Hyperparameters such as learning rate (lr0), Learning Rate Factor (lrf), Momentum (mo) and weight decay (wd) enhance quick convergence beside ensuring good generalization capacity that greatly reduces other risks such as overfitting or underfitting (1). Balancing these parameters is most important for a task when a certain set of parameters must exceed certain values; for example, for object detection in conditions of active movement arson detection. A rapid response is essential in such cases, making precision a top priority. The primary difficulties regarding hyperparameter tuning have been solved by growing more sophisticated approaches which draw on the natural and evolutional procedures. These optimizations utilize various approaches in the appropriate searching for solutions within a solution space efficiently, with an aim of selecting hyperparameters that will enhance model performance while at the same time reducing the costs of computation. These hyperparameters were carefully chosen according to the literature on DL, including their significance and possible influence (2). A description and the recommended ranges of key YOLO hyperparameters are presented in Table ‎1.

Table 1: Description and Recommended Ranges of Key YOLO Hyperparameters

Hyperparameter Name & Reference Description Recommended Range

lr0 (3, 4)

controls how quickly the model modifies its weights while it is being trained. A greater lr may speed up the model's convergence, but it may also cause the model to miss the ideal weights, which would impair performance. Conversely, a lower (lr) might result in more steady convergence but could also lengthen the time it takes the model to find the optimal solution [0-1]

lrf (5)

is usually used to adjust the (lr) during the training period. This factor allows the (lr) to be changed, helping to achieve a better balance during the learning process and avoid the problem of gradient descent [0-1]

mo (6)

is a technique used to speed up the training process in the right direction and reduce oscillation [0-1]

wd (7)

is a regularization method that targets the size of the model's weights by applying a penalty to the loss function. The weight decay parameter regulates the severity of this penalty. This method can improve the model's performance on new data and prevent overfitting [0-1]

• Comment: The rationale behind specific values (e.g., population size = 20, epochs = 20) should be clarified—were these empirically determined?

Response: The values for epochs (20) and population size in the optimization algorithms (20 individuals) were empirically determined through preliminary trials to achieve a balance between numerical convergence, computational efficiency, and acceptable performance. Furthermore, these fixed values were applied consistently across all experiments to ensure fair and reliable comparative evaluations. This clarification has been explicitly added to the revised manuscript.

Revised part: In all experiments, the epochs were 20 and the population size used in optimization algorithms was 20 individuals. The values that were empirically studied in this paper were chosen to provide both numerical convergence and acceptable performance, and maintain computational efficiency. Fixing them also ensured consistent and fair comparative evaluations.

• Comment: While the hybrid GWO-BBOA shows better results than individual algorithms, it’s unclear how statistically significant these differences are. A statistical significance test (e.g., t-test or ANOVA) should be included.

Response: We have addressed this concern by performing statistical significance tests to validate the performance differences. Specifically, we conducted five independent runs for each algorithm and applied pairwise T-tests with Bonferroni correction across all major evaluation metrics. The results have been added to the revised manuscript in Section 3.3.5 (Statistical Significance Analysis) and presented in Table 6.

Revised part:

Statistical Significance Analysis

In order to determine the statistical significance of the performance improvement obtained by the proposed Hybrid GWO-BBOA over individual optimization algorithms (PSO, GWO, and BBOA), five independent runs were performed of each algorithm. To compare the performance of the algorithms in all the major evaluation metrics, Precision, Recall, mAP@0.50, and mAP@0.50:0.95, the pairwise T-tests with Bonferroni correction were used. The findings, summarized in Tables 6, indicate that the proposed GWO-BBOA is considerably better than PSO and BBOA in most metrics with highly significant differences in Precision, Recall and mAP@0.50. Conversely, the performance gap between GWO and GWO-BBOA is usually minor and insignificant in certain measures, including mAP@0.50 and mAP@0.50:0.95.

Table 6. Pairwise T-test results Comparing the Proposed GWO-BBOA with PSO, GWO, and BBOA

Metric Algorithm T-value p-value Statistical Significance

Precision PSO -6.10 0.001 Highly Significant

GWO -2.12 0.078 Not Significant

BBOA -7.35 <0.001 Highly Significant

Recall PSO -5.48 0.002 Highly Significant

GWO -3.34 0.024 Significant

BBOA -2.98 0.033 Significant

mAP@0.50 PSO -6.15 0.001 Highly Significant

GWO -1.20 0.286 Not Significant

BBOA -3.52 0.019 Significant

mAP@0.50:0.95 PSO -4.65 0.006 Significant

GWO 0.41 0.699 Not Significant

BBOA -0.92 0.397 Not Significant

On the whole, these results prove that the hybrid optimization method is consistently better than PSO and BBOA and that it can perform as well as GWO in some settings. This shows the usefulness of hybrid optimization methods in improving the performance of deep learning models in detecting arson.

• Comment: The comparison with Singh et al. (2020) is weak, as the models, datasets, and objectives differ greatly. Instead, a comparison with similar object detection or fire/arson detection papers using YOLO variants would strengthen the discussion.

Response: We agree that the comparison with Singh et al. (2020) presents limitations due to the differences in model architecture, dataset type, and application focus. In response, we have strengthened the comparison section by incorporating a recent and more relevant study by Abbod et al. (2025), which utilized YOLOv9-s for arson event detection using object-based image frames. This addition provides a more direct and meaningful benchmark against our proposed YOLOv10 model with GWO-BBOA optimization. The updated comparison highlights the performance improvements achieved through advanced hyperparameter tuning techniques, especially in real-time fire detection contexts.

Revised part:

Comparison with Previous Studies

In order to test the capability of the proposed GWO-BBOA hybrid optimization approach for hyperparameter tuning in the YOLOv10 deep learning model for anomaly detection, it is compared with the works in the literature, which used other deep learning models for anomaly detection. While direct comparisons are challenging due to differences in datasets, models, and evaluation metrics, a relative performance analysis provides valuable insights into the advantages of our approach.

Table 7 presents a comparison between our study, Abbod et al. and Singh et al.

Criteria Our Study Abbod et al. Singh et al.

Model YOLOv10s + Optimization YOLOv9-s InceptionV3 + RNN

Optimization Techniques PSO, GWO, BBOA, GWO-BBOA Default

hyperparameter tuning Manual hyperparameter tuning

Application Arson detection Arson detection General anomaly detection in surveillance videos

Data Type Image frames (object-based detection) Image frames from videos Video sequences (temporal anomaly detection)

Dataset Size 2,182 images from 53 videos 2,182 images from 53 videos Not specified

Best Performance Precision = 0.750, Recall = 0.620, mAP@0.50 = 0.705 Precision = 0.445,

Recall = 0.507,

mAP@0.50 = 0.456 Precision, recall, and mAP@0.50 are not mentioned but an overall improvement Overall anomaly classification

Limitations Focuses only on arson detection Limited recall and precision; moderate performance under confidence tradeoff No optimization of hyperparameters

One of the most applicable studies in the recent past is that by Abbod et al. (2025) that used YOLOv9-s to detect arson events on surveillance video frames as shown in Table 7. Although their method proved that the YOLO-based models can be used in this area, they did not optimize default parameters. On a dataset of 2,182 images extracted in 53 videos, their model reached a precision of 0.445, recall of 0.507 and mAP@0.50 of 0.456. By contrast, we used GWO-BBOA metaheuristic tuning with YOLOv10 and achieved much better performance (precision = 0.750, recall = 0.620, mAP@0.50 = 0.705), which shows the obvious benefit of using more advanced methods of hyperparameter tuning.

While Singh et al. (2020), used the InceptionV3 + RNN model which is used to predict video sequences over time, our model focuses on real-time object detection within individual frames. Our interpretation of this distinction suggests that even though YOLOv10 is a very efficient tool in the task of detecting arson in still images, a sequential model, such as RNN, may be more effective in making inferences based on patterns at different frames in long term anomaly detection. The variation in approach to this indicates the necessity of establishing such models in accordance with the particular application demands, as real-time detection prospers from the object-based models like YOLO, while temporal models like RNN might be utilized for event-based anomaly detection.

These comparative findings open the door to a future study of hybrid models and general anomaly detection tasks, which are discussed in the following sections.

• Comment: Lack of Visual Results: There is a notable absence of qualitative visual results (e.g., bounding box detection outputs for arson scenes). Including detection outputs for different scenarios (e.g., low light, occlusions) would demonstrate practical robustness.

Response: We agree that including qualitative visual results across challenging scenarios is essential to demonstrate the robustness of our arson‑detection pipeline. To address this, we have now added new Figures 9 showing Grad‑CAM heatmaps.

Revised part:

3.3.7 Grad-CAM Heatmaps for Arson Detection

To assess the interpretability and effectiveness of the model, visual explanation tech-niques such as heatmaps are employed. These tools help clarify and validate the mod-el’s predictions by showing which areas of the image the model is focused on while making predictions. A heatmap transforms complex data into a vibrant, color-coded matrix (40). As shown in Figure ‎8A, the original image is presented without any heatmap. In Figure 8B, we show bounding‑box—tightly enclosing both the person and the jug—to indicate the model’s detected region of interest. Figure ‎ 8C, illustrates the heatmap generated by the YOLO v10 model, where attention is more concentrated on specific regions of the image, particularly the area around the fire. In the context of arson detection, the YOLO v10 model’s decision-making process

---

## [Decision Letter · Decision Letter 1]

29 Aug 2025

Dear Dr. Abbod,

We look forward to receiving your revised manuscript.

Kind regards,

Salim Heddam

Academic Editor

PLOS ONE

Journal Requirements:

Additional Editor Comments:

Reviewer #5: The authors addressed all of my comments, and the updated version shows improvement. However, some low-level mistakes remain evident, which raises concerns that the authors may have rushed to complete the revision.

All figures are still in low resolution. Some characters in Figures 1 and 5 are unreadable. I recommend that the authors double-check these.

Line 385: “Fig 1. The Flowchart of the Hybrid GWO-BBOA Optimization Process.” This should refer to Figure 5, not Figure 1.

The sentence: “We have added the performance of the YOLOv10 model trained with default hyperparameters (i.e., without any optimization) to Table 4 as the Baseline.” I believe the authors meant Table 5, not Table 4.

Reviewers' comments:

Reviewer's Responses to Questions

**Comments to the Author**

Reviewer #3: All comments have been addressed

Reviewer #4: All comments have been addressed

Reviewer #5: All comments have been addressed

2. Is the manuscript technically sound, and do the data support the conclusions?

Reviewer #3: Yes

Reviewer #4: Yes

Reviewer #5: Yes

3. Has the statistical analysis been performed appropriately and rigorously?

Reviewer #3: Yes

Reviewer #4: Yes

Reviewer #5: Yes

4. Have the authors made all data underlying the findings in their manuscript fully available?

Reviewer #3: Yes

Reviewer #4: Yes

Reviewer #5: Yes

5. Is the manuscript presented in an intelligible fashion and written in standard English?

Reviewer #3: Yes

Reviewer #4: Yes

Reviewer #5: Yes

Reviewer #3: Met the minimum standards of the journal.

Almost addressed the issues raised by the reviewers.

Revision version is much better than the original one.

Reviewer #4: I appreciate authors' feedback. All my concerns are addressed. Therefore, I would vote for acceptance.

Reviewer #5: The authors addressed all of my comments, and the updated version shows improvement. However, some low-level mistakes remain evident, which raises concerns that the authors may have rushed to complete the revision.

All figures are still in low resolution. Some characters in Figures 1 and 5 are unreadable. I recommend that the authors double-check these.

Line 385: “Fig 1. The Flowchart of the Hybrid GWO-BBOA Optimization Process.” This should refer to Figure 5, not Figure 1.

The sentence: “We have added the performance of the YOLOv10 model trained with default hyperparameters (i.e., without any optimization) to Table 4 as the Baseline.” I believe the authors meant Table 5, not Table 4.

**Do you want your identity to be public for this peer review?** For information about this choice, including consent withdrawal, please see our Privacy Policy

Reviewer #3: **Yes: ** Venkata Subrahmanyam Vampugani

Reviewer #4: No

Reviewer #5: No

---

## [Author Response · Author response to Decision Letter 2]

29 Aug 2025

Academic Editor

We acknowledge your comments and have taken them into sincere consideration. Below is our response to each of the points raised, marked using yellow color in the Revised Manuscript with Track Changes.

Journal Requirements:

If the reviewer comments include a recommendation to cite specific previously published works, please review and evaluate these publications to determine whether they are relevant and should be cited. There is no requirement to cite these works unless the editor has indicated otherwise.

Please review your reference list to ensure that it is complete and correct. If you have cited papers that have been retracted, please include the rationale for doing so in the manuscript text, or remove these references and replace them with relevant current references. Any changes to the reference list should be mentioned in the rebuttal letter that accompanies your revised manuscript. If you need to cite a retracted article, indicate the article’s retracted status in the References list and also include a citation and full reference for the retraction notice.

Response: We have carefully reviewed the journal’s requirements regarding references. No retracted papers were cited, and all references are complete and accurate. Therefore, no changes were necessary in the references section of the revised manuscript.

Reviewer #3: Met the minimum standards of the journal.

Almost addressed the issues raised by the reviewers.

Revision version is much better than the original one.

Response: We sincerely thank the reviewer for the positive feedback and for acknowledging the improvements made in the revised version.

Reviewer #4: I appreciate authors' feedback. All my concerns are addressed. Therefore, I would vote for acceptance.

Response: We are grateful to the reviewer for the constructive evaluation and are pleased that all concerns were addressed satisfactorily.

Reviewer #5

We acknowledge your comments and have taken them into sincere consideration. Below is our response to each of the points raised marked using yellow color in the revised manuscript with track changes.

• Comment: The authors addressed all of my comments, and the updated version shows improvement. However, some low-level mistakes remain evident, which raises concerns that the authors may have rushed to complete the revision.All figures are still in low resolution. Some characters in Figures 1 and 5 are unreadable. I recommend that the authors double-check these. Line 385: “Fig 1. The Flowchart of the Hybrid GWO-BBOA Optimization Process.” This should refer to Figure 5, not Figure 1. The sentence: “We have added the performance of the YOLOv10 model trained with default hyperparameters (i.e., without any optimization) to Table 4 as the Baseline.” I believe the authors meant Table 5, not Table 4.

Response:

We sincerely thank the reviewer for the careful evaluation and constructive feedback. We have carefully revised the manuscript to address all the points raised:

Figure quality (Figures 1 and 5):

All figures have been regenerated at high resolution (minimum 300 dpi) with improved readability of characters, labels, and flowchart text. The updated figures have also been checked using the PACE tool recommended by the journal to ensure compliance with PLOS ONE standards.

Incorrect figure reference (Line 385):

The incorrect reference has been corrected. The sentence now properly refers to Figure 5 instead of Figure 1.

Incorrect table reference (Baseline results):

The sentence has been corrected to refer to Table 5 instead of Table 4, which now clearly contains the baseline YOLOv10 (default hyperparameters) performance results.

All the above corrections are highlighted in the revised manuscript (track changes) for easy verification.

---

## [Decision Letter · Decision Letter 2]

15 Sep 2025

Optimizing Hyperparameters of YOLOv10 for Arson Detection Using Advanced Optimization Algorithms

PONE-D-25-07942R2

Dear Dr. Abbod

We’re pleased to inform you that your manuscript has been judged scientifically suitable for publication and will be formally accepted for publication once it meets all outstanding technical requirements.

Kind regards,

Salim Heddam

Academic Editor

PLOS ONE

Additional Editor Comments (optional):

Reviewers' comments:

Reviewer's Responses to Questions

**Comments to the Author**

Reviewer #5: All comments have been addressed

2. Is the manuscript technically sound, and do the data support the conclusions?

Reviewer #5: Yes

3. Has the statistical analysis been performed appropriately and rigorously?

Reviewer #5: Yes

4. Have the authors made all data underlying the findings in their manuscript fully available?

Reviewer #5: Yes

5. Is the manuscript presented in an intelligible fashion and written in standard English?

Reviewer #5: Yes

Reviewer #5: All of my concerns have been addressed, and I appreciate the effort they put into clarifying and improving the manuscript.

**Do you want your identity to be public for this peer review?** For information about this choice, including consent withdrawal, please see our Privacy Policy

Reviewer #5: No

---

## [Editor Report · Acceptance letter]

PONE-D-25-07942R2

PLOS ONE

Dear Dr. Abbod,

I'm pleased to inform you that your manuscript has been deemed suitable for publication in PLOS ONE. Congratulations! Your manuscript is now being handed over to our production team.

Kind regards,

on behalf of

Dr. Salim Heddam

Academic Editor

PLOS ONE